

# Landscape reconstructions for Europe during the late Last Glacial (60–20 ka BP): A pollen-based REVEALS approach

Oliver A. Kern[1], Andreas Maier[2], Nikki Vercauteren[1]

[1]Institute of Geophysics and Meteorology, University of Cologne, Pohligstr. 3, 50969 Cologne, Germany

[2]Institute of Prehistoric Archaeology, University of Cologne, Bernhard-Feilchenfeld-Str. 11, 50969 Cologne, Germany

*Correspondence to*: Oliver A. Kern (okern@uni-koeln.de)

**Abstract.** Vegetation change during the Last Glacial period in Europe plays a crucial role in better understanding the ecosystem
dynamics response to abrupt climate change. Yet, quantitative reconstructions of land-cover primarily focus on the Holocene
period and aim to disentangle the impact of anthropogenic and climatic stress on the vegetation. Here, we present temporally
continuous land-cover reconstructions from Europe for the latter half of the Last Glacial period (60–20 ka BP) using the
"Regional Estimates of Vegetation Abundance from Large Sites" (REVEALS) model. The pollen-based REVEALS model
uniquely factors in plant-specific parameters, such as relative pollen productivity (RPP) and pollen fall speed to model pollen
dispersal and thus provides more accurate representation of past vegetation cover than fossil pollen data. We compiled a total
of 61 datasets from Europe and its bordering regions to model land-cover estimates across 60 time steps in 1000 year
increments. By grouping the 38 analysed taxa into 5 land-cover types (LCTs), we simplify the interpretation of our results and
demonstrate this using three crucial periods during the Last Glacial: Greenland Interstadial 14 (GI-14), Greenland Stadial 9
(GS-9), and the Last Glacial Maximum (LGM). These periods provide insight into stadial-interstadial vegetation variability as
well as extreme glacial conditions, which seem to play a fundamental role in the demographic developments of Palaeolithic
hunter-gatherers. Additionally, we compare the REVEALS land-cover estimates to raw pollen data and also provide
REVEALS standard errors and discuss the reliability of our results as well as potential avenues to further improve the reliability
of REVEALS estimates. To facilitate the use and interpretation of our data for a wide scientific audience, we developed the
browser-based application PALVEG (https://oakern.shinyapps.io/PALVEG/), which requires no prior programming
experience and dynamically generates maps based on user input.



## 1 Introduction

The quantification of terrestrial vegetation dynamics in response to climatic change remains poorly understood as it
necessitates the conversion of fossil pollen data to tangible reconstructions of the past vegetation cover (Dallmeyer et al.,
2023). Yet, terrestrial vegetation is one of the major components of the Earth's terrestrial biosphere and a major driver in the
global carbon cycle (Masson-Delmotte et al., 2021). Through biogeochemical feedback processes, their role as a mitigation
agent is of particular importance in the context of climate change (Anderegg et al., 2020; Harper et al., 2018; Williamson,
2016). Making sense of the past and developing future adaptation and mitigation strategies greatly depend on our understanding
of the interaction between climate and vegetation – an integral part of Earth system models (ESMs) employed to generate
climate projections (Fisher et al., 2018; McDowell et al., 2020). ESMs often rely on dynamic global vegetation models
(DGVMs) that predict changes in land-cover, productivity, biomass, and/or carbon storage as a function of static bioclimatic
variables that are provided by climate models (e.g., Shao et al., 2018; Lu et al., 2018; Hopcroft et al., 2017). Evaluating and
improving DGVMs requires appropriate data.

An ongoing endeavour is the comparison of DGVM outputs with palaeoproxy records of the past vegetation composition
(Shao et al., 2018; Woillez et al., 2013). The primary goals are to evaluate the output DGVMs and, in a next step, to inform
DGVMs and thereby improve the accuracy of their predictions. Pollen records from peat deposits, lake sediments, marine
sediments, and to a lesser extent from outcrops and archaeological layers represent the most promising empirical data of past
vegetation cover (Dallmeyer et al., 2023). Pollen records provide direct evidence of the vegetational composition and are
widespread across terrestrial landscapes. However, the pollen-vegetation relationship is non-linear and often biased by taxon-
specific characteristics that influence pollen productivity, dispersal, and deposition (Andersen, 1970; Prentice, 1985).
Moreover, boundary conditions, such as $CO_2$-levels, atmospheric patterns, and soil properties, vary through time and may
influence these taxon-specific parameters (Ladeau and Clark, 2006; Wayne et al., 2002) thus hindering a direct comparison
with DGVM outputs.

Despite these proxy-specific drawbacks, quantitative vegetation reconstructions have a long history in the geosciences
(Chevalier et al., 2020). Over time, different methods have emerged and improved upon, such as biomisation (e.g., Prentice et
al., 1996; Binney et al., 2017), the Pseudobiomisation Method (PBM; Fyfe et al., 2010; Woodbridge et al., 2014), modern
analogue techniques (MATs; e.g., Zanon et al., 2018), and landscape reconstructions (REVEALS; Sugita, 2007; Githumbi et
al., 2022; Serge et al., 2023; Schild et al., 2024). The different approaches have been discussed, compared, and evaluated
previously (Chevalier et al., 2020). Here, we focus on the Regional Estimates of Vegetation Abundance from Large Sites
(REVEALS; Sugita, 2007). Among the aforementioned techniques, REVEALS uniquely factors in pollen productivity and
dispersal properties, yielding land-cover percentages for each individual taxon. Syntheses of regional to continental
REVEALS-specific parameters are continually expanded and improved upon (e.g., Githumbi et al., 2022; Mazier et al., 2012;
Schild et al., 2024; Serge et al., 2023; Wieczorek and Herzschuh, 2020). In the recent years, REVEALS land-cover simulations
were generated on global (Schild et al., 2024) and continental scales for Eurasia, China, and North America (e.g., Roberts et



al., 2018; Githumbi et al., 2022; Li et al., 2023). However, previous works have mainly focused on the response of terrestrial ecosystems to anthropogenic stress (e.g., agriculture and deforestation) during the Holocene period (11,700 yrs BP to present).

**Figure 1: Top: Map of Europe depicting the location of all pollen archives included in this study. See Appendix A (Tab. A1) for a detailed legend and references. Last Glacial Maximum ice-sheet extent redrawn after Ehlers et al. (2011). Bottom: Greenland oxygen isotopes (Rasmussen et al., 2014) for the last ca. 125,000 years. Red intervals are referred to in the results and discussion section (section 3). Approximated presence of different human species (Staubwasser et al., 2018) is indicated by the coloured bars. Marine Isotope Stages (MIS) after (Lisiecki and Raymo, 2005).**



The Holocene is a period of relative climate stability and low-amplitude climate fluctuations in comparison with the preceding glacial period (Mayewski et al., 2004). In order to focus on the impact of climatic stressors on the regional vegetation, we direct our attention on the Last Glacial period (ca. 115,000–11,700 yrs BP), which features marked climatic fluctuations (Fig. 1) with limited human impact on the environment (Brovkin et al., 2021; Fletcher et al., 2010). The Last Glacial encompasses a wide range of climate states: Following the relatively stable interglacial climates of early Marine Isotope Stage 5 (MIS 5e), phases of climate instability (remainder of MIS 5a–d and MIS 3) alternated with relatively stable glacial climates (MIS 4 and MIS 2; Fig. 1). Particularly the climate in the Northern Hemisphere during MIS 3 is characterised by a marked climate instability that manifested in several periods of abrupt warming that permeated the otherwise glacial climatic conditions (e.g., Rasmussen et al., 2014; Corrick et al., 2020). These periods were termed Greenland Interstadials (GIs) after their discovery in Greenland ice cores (Svensson et al., 2006), while periods in-between GIs are termed Greenland Stadials (GSs), accordingly. While these events are relatively well-studied in the marine realm, their impact on terrestrial palaeoenvironments is still largely unknown (Fletcher et al., 2010; Moreno et al., 2014), especially in Central Europe (Britzius et al., 2024; Kern et al., 2022; Woillard, 1978), a fact that is commonly ascribed to the scarcity of available data. Although the climatic boundary conditions today are different, the response of terrestrial ecosystems to abrupt warming events is of particular interest in the context of modern climate change (Masson-Delmotte et al., 2021). The vegetation response to GI-GS variability raises several questions regarding adaptation, resilience, and extinction/repopulation rates (Alley et al., 2003; O'Neill et al., 2017), the possibility of local/regional plant refugia (Tzedakis et al., 2013; Willis et al., 2000; Willis and van Andel, 2004), and also seems to have high relevance for the large-scale demographic developments of Palaeolithic hunter-gatherers (Maier et al., 2022).

## 2 Methods

### 2.1 Pollen data – sources and preparation

A total of 61 pollen records with data for the 60–20 kyr BP interval were compiled from study sites located in Europe and bordering regions (Fig. 1). The majority of those datasets were obtained from the Neotoma database (Williams et al., 2018), the ACER pollen and charcoal database (Sánchez Goñi et al., 2017), and PANGAEA (Felden et al., 2023). Additional datasets were gathered directly from the respective publications and by personal communication with other researchers. A complete table listing all sites, data sources, and references can be found in Appendix A (Tab. A1).

Most datasets were either provided with a robust age-depth model in the corresponding publication or database. Records without a robust chronology were tentatively correlated with well-established records based on the stratigraphic placement of these records in a broader context. Datasets from marine records were excluded due to their overrepresentation of *Pinus* pollen grains (Heusser and Balsam, 1977). For the purpose of this study, raw pollen counts (compared to e.g., pollen percentages) are required (see section 2.3 for details). However, some datasets only provide pollen percentages and the original data could not be obtained. To maximise the amount of records included in the study, percentage data from these sites were recalculated to a

baseline counting sum of 300, which is considered the minimum for statistically robust results and thus often used as a target during palynological analysis (Weng et al., 2006).

Data preparation included the removal of empty and duplicate entries and excluding samples outside of the desired age interval to speed up the REVEALS calculation. Subsequently, all data were aggregated into 1000-year time bins following Githumbi
et al. (2022) and Trondman et al. (2016). Since not all pollen records cover the entire target interval, the availability of suitable datasets for each time bin varies with time (see Appendix B, Fig. B1). Data aggregation reduced the input dataset from 5231 to 1632 samples. Afterwards, the harmonisation of nomenclature and pollen morphological types for all datasets was required (Birks et al., 2023). Taxa harmonisation was performed in R using and adapting the taxon harmonisation table supplied by Githumbi et al. (2022).

## 2.2 REVEALS

The REVEALS model is based on the r-value model (Davis, 1963) and subsequent iterations (Andersen, 1970; Parsons and Prentice, 1981) and its development is described in detail in earlier studies (e.g., Sugita, 2007). In essence, REVEALS reconstructs land-cover estimates for all taxa, factoring in pollen counts, basin parameters, such as basin type (i.e., lake or bog) and basin radius (assuming a circular basin), and plant physiological parameters (pollen productivity and pollen fall speed).
Early versions of REVEALS were limited to large lakes (>50 ha) based on the premise that sediments from large lakes accurately represent regional vegetation assemblages (Sugita, 2007). Subsequent development stages of REVEALS then successfully demonstrated that REVEALS estimates from small lakes (<50 ha) and bogs (both small and large) are comparable to those of a large lake in the same region (Trondman et al., 2016). The substantially higher standard deviation for these data can be offset by including multiple small sites, which is consequently highly encouraged (Trondman et al., 2016).

## 2.3 REVEALS input parameters

The REVEALS model estimates land-cover percentages using raw pollen counts from palaeoclimate archives, supplied on either a depth or an age scale. In addition to pollen data, the REVEALS model requires additional taxa-specific parameters, such as the relative pollen productivity (RPP) estimates and the respective standard deviation as well as the fall speed of pollen (FSP). RPP and FSP values have been previously compiled on continental to global scales (Githumbi et al., 2022; Serge et al.,
2023; Wieczorek and Herzschuh, 2020). Moreover, REVEALS land-cover estimates using variable RPP values have been previously validated and optimized through comparison with satellite-based land-cover reconstructions (Hansen et al., 2013; Schild et al., 2024). Here, we use combined RPP and FSP values from Githumbi et al. (2022) and Serge et al. (2023) that are optimized for Europe, but briefly also discuss differences in land-cover estimates by using others sets of RPP and FSP (Tab. 1). For maximum comparability, we use the same taxa across for all model runs and fill missing values as needed from
Githumbi et al. (2022) and Serge et al. (2023).





**Table 1: Fall speed (FSP), relative pollen productivity (RPP) including standard deviation (SD) and associated land-cover type (LCT) for all taxa included in this study. We list three sets of RPP values: A synthesis of RPP values for Europe, modified from Githumbi et al. (2022) and Serge et al. (2023); a global compilation (listed are values for Europe, or, if not available, the Northern Hemisphere; Wieczorek and Herzschuh, 2020, dataset v2), and optimized RPP values using a satellite-based validation and optimization routine (Schild et al., 2024). All values for RPP and SD (RPP) are relative to Poaceae.**

| | this paper, synthesis after Githumbi et al. (2022) & Serge et al. (2023) | | Wieczorek & Herzschuh (2020) dataset v2 | | Schild et al. (2024) | |
| **Taxa** | **FSP** | **RPP (SD)** | **FSP** | **RPP (SD)** | **optimized RPP** | **Land-cover type (LCT)** |
| --- | --- | --- | --- | --- | --- | --- |
| *Abies alba* | 0.120 | 6.875 ± 1.442 | 0.120 | | | |
| *Juniperus communis* | 0.016 | 2.070 ± 0.040 | 0.016 | 7.940 ± 1.280 | | **Conifers** |
| *Picea abies* | 0.056 | 5.437 ± 0.097 | 0.056 | 1.650 ± 0.150 | 6.490 | |
| *Pinus* | 0.031 | 6.058 ± 0.237 | 0.036 | 10.860 ± 0.800 | 43.440 | |
| *Alnus* | 0.021 | 13.562 ± 0.293 | 0.021 | 8.490 ± 0.220 | 2.124 | **Cold deciduous trees** |
| *Betula* | 0.024 | 5.106 ± 0.303 | 0.024 | 4.940 ± 0.440 | 19.759 | |
| *Salix* | 0.022 | 1.182 ± 0.077 | 0.028 | 0.390 ± 0.060 | | |
| *Acer* | 0.056 | 0.800 ± 0.230 | 0.056 | 0.230 ± 0.040 | | |
| *Buxus sempervirens* | 0.032 | 1.890 ± 0.068 | 0.032 | | | |
| *Carpinus betulus* | 0.042 | 4.520 ± 0.425 | 0.042 | 3.090 ± 0.284 | | |
| *Corylus avellana* | 0.025 | 1.710 ± 0.100 | 0.025 | | 0.831 | |
| *Fagus sylvatica* | 0.057 | 5.863 ± 0.176 | 0.056 | 2.350 ± 0.107 | 0.759 | **Temperate deciduous trees** |
| *Fraxinus excelsior* | 0.022 | 1.044 ± 0.048 | 0.022 | 2.970 ± 0.250 | | |
| *Populus* | 0.025 | 2.660 ± 1.250 | 0.025 | 3.420 ± 1.600 | | |
| *Quercus* deciduous | 0.035 | 4.537 ± 0.086 | 0.035 | 2.920 ± 0.100 | 11.601 | |
| *Tilia* | 0.032 | 1.210 ± 0.116 | 0.032 | 0.930 ± 0.090 | | |
| *Ulmus* | 0.032 | 1.270 ± 0.050 | 0.026 | 2.240 ± 0.462 | | |
| *Carpinus orientalis* | 0.042 | 0.240 ± 0.070 | 0.042 | 3.090 ± 0.284 | | |
| *Castanea sativa* | 0.010 | 3.258 ± 0.059 | 0.014 | 5.870 ± 0.245 | | |
| *Phillyrea* | 0.015 | 0.512 ± 0.076 | 0.015 | | | **Mediterranean trees** |
| *Pistacia* | 0.030 | 0.755 ± 0.201 | 0.030 | | | |
| *Quercus* evergreen | 0.015 | 11.043 ± 0.261 | 0.015 | | | |
| *Sambucus nigra* | 0.013 | 1.300 ± 0.116 | 0.013 | | | |
| Apiaceae | 0.042 | 0.260 ± 0.010 | 0.042 | 2.130 ± 0.410 | | |
| *Artemisia* | 0.025 | 3.937 ± 0.146 | 0.014 | 4.330 ± 1.590 | | |
| Asteraceae | 0.051 | 0.360 ± 0.137 | 0.032 | 0.220 ± 0.020 | 0.055 | |
| Chenopodiaceae | 0.019 | 4.280 ± 0.270 | 0.019 | 4.280 ± 0.270 | | |
| Cyperaceae | 0.035 | 0.962 ± 0.050 | 0.035 | 0.560 ± 0.020 | 0.188 | |
| Ericaceae | 0.051 | 0.070 ± 0.040 | 0.030 | 0.440 ± 0.020 | 0.109 | |
| Fabaceae | 0.021 | 0.400 ± 0.070 | 0.021 | 0.400 ± 0.070 | | |
| *Filipendula* | 0.006 | 3.000 ± 0.285 | 0.006 | 0.530 ± 0.050 | | **Open Land** |
| *Plantago* | 0.029 | 1.447 ± 0.170 | 0.028 | 2.490 ± 0.110 | | |
| Poaceae | 0.035 | **1.000 ± 0.000** | 0.035 | **1.000 ± 0.000** | | |
| *Potentilla* | 0.018 | 1.190 ± 0.133 | 0.018 | 0.530 ± 0.050 | | |
| Ranunculaceae | 0.014 | 1.960 ± 0.360 | 0.014 | 0.990 ± 0.120 | | |
| Rubiaceae | 0.019 | 3.950 ± 0.314 | 0.019 | 1.560 ± 0.120 | | |
| *Rumex acetosa* | 0.018 | 3.020 ± 0.278 | 0.018 | 0.580 ± 0.030 | | |
| *Urtica* | 0.007 | 10.520 ± 0.310 | 0.007 | 10.520 ± 0.310 | | |

The REVEALS model has previously been used to estimate anthropogenic land-cover change during the Holocene and includes taxa of significance for this specific research question, such as *Cerealia* (cereals) and *Secale* (rye), which are indicators of agricultural practices. Since agriculture only emerged during the early to mid-Holocene in Europe (Price, 2000;

Zeder, 2011), we grouped these taxa together with other Poaceae (grasses) in our analysis. *Plantago lanceolata* (plantain), while also a common indicator for agricultural practices, is grouped together with other taxa from the genus *Plantago*, as pollen from this family naturally occur in pollen records from Europe throughout the Last Glacial period. The respective RPP value

is calculated as the mean of the available values for different *Plantago* species (Githumbi et al., 2022).

Pollen data from different archives are typically heterogeneous in terms of nomenclature and the level at which taxa are identified. Tree taxa are most commonly identified on the genus (e.g., *Picea*) or species level (e.g., *Picea abies*). However, herbs and grasses are typically identified on the family level (e.g., Ericaceae and Poaceae), owing to the sheer diversity of these groups (the Poaceae family contains more than 10,000 species in over 700 genera), rendering the exact identification

using microscopic analysis almost impossible. To include as many data as possible, it was necessary to harmonise the pollen data. Given the limited number of datasets to begin with, we resorted to simplifying all datasets to the most common denominator (family level) rather than excluding datasets to overcome the heterogeneity in pollen data. For some taxa, RPP values for boreal/temperate and Mediterranean specimens exist (Githumbi et al., 2022). Since we focus on the Last Glacial, i.e., a period when mostly cold and dry climates prevailed in Europe, we chose the boreal/temperate values, as we expect

Mediterranean taxa to play a subordinate role under glacial conditions compared to their boreal/temperate counterparts. As a result, our final RPP and FSP table consists of entries for 38 taxa (Tab. 1).

Additional parameters used by REVEALS are basin radius (in m), basin type (i.e., lake or bog), and the geographical location of the archive (in latitudinal and longitudinal coordinates). This information was either taken directly from the respective publications, included in database metadata, or roughly estimated using satellite imagery. During the Last Glacial, peat bogs

were much less common than during the Holocene and many lakes transformed into bogs as the climatic conditions became warmer and wetter (e.g., Füramoos, Germany). Hence, we set the basin type to "lake" for all sites with no information, unless the respective publications indicated that a bog was present throughout most of the history of the archive (e.g., Tenaghi Philippon, Greece).

REVEALS also requires two independent parameters: wind speed and the maximum extent of the regional vegetation (Zmax).

Following a previous evaluation of REVEALS estimates combined with empirical data, wind speed and Zmax were set to 3 m/s and 100 km, respectively (Trondman et al., 2015).

## 2.4 Implementation of REVEALS

The REVEALS model was implemented using the R package LRA (Abraham et al., 2014). The REVEALS function within LRA distinguishes between data from lakes and bogs and accounts for the taxon-specific FSP and RPP (including standard

deviation). LRA offers two different dispersal models: a Gaussian plume model (used in this study; Sutton, 1953) and a Lagrangian stochastic model (Theuerkauf et al., 2016).

Previous studies have mostly implemented a grid-based approach to combine records across 1° x 1° grid cells in Europe for a combined mean cover estimate and standard error for all taxa. Owing to the scarcity of datasets available in Europe during the pre-Holocene, such an approach is not feasible, as the typical distance between two pollen archives exceeds 1° in either

direction. Hence, we chose not to apply this technique to our dataset and provide site-specific estimates, rather than regional (grid-cell wide) estimates. To provide maximum clarity, we specify which sites are considered "small" (i.e., < 50 ha) and advocate for a careful interpretation of REVEALS estimates from these sites.

The REVEALS output generates land-cover estimates (including standard errors) for all taxa from all samples provided. For the purpose of this study, we provide REVEALS estimates from all sites in 1000-year increments, with a ± 500-year data

aggregation window to account for age-depth model uncertainties. As a result, some sites may not be present during all time slices, depending on their temporal resolution and/or potential gaps in the respective datasets (see Figs. 3–5).

### 2.5 Land Cover Types

In the biomisation approach (e.g., Prentice et al., 1996; Binney et al., 2017; Bigelow et al., 2003) each taxon is assigned a plant functional type (PFTs), based on their ecological and climatic preferences. Subsequently, groups of PFTs form different

biomes, such as desert, steppe, various types of tundra and forests. To simplify interpretation, PFTs can also be grouped into "Super-PFTs" (Binney et al., 2017) as an alternative to biomes. The REVEALS model only includes a limited number of taxa due to the limited availability of reliable RPP estimates. Therefore, we adapted a simplified approach introduced by Githumbi et al. (2022) and group taxa into five land-cover types (LCTs): Open land (which includes all types of grasses and herbaceous taxa), conifers, cold deciduous trees, temperate deciduous trees, and Mediterranean trees. Additionally, we provide estimates

for total trees, calculated as 100 % - Open land. See Appendix C (Tab. C1) for a detailed list of all taxa included in each LCT. The choice of LCTs is based on the vegetational composition during the glacial conditions of the Late Pleistocene and aims to simplify and illustrate broader changes in the vegetation rather than highlight a single taxon.

### 2.6 Mapping

The results of the REVEALS land-cover reconstructions are presented as maps in time slices of 1000-year steps for all taxa

and all LCTs. Figures 3-5 illustrate exemplary maps during full glacial (LGM, 23 ka BP), interstadial (GI-14, 52 ka BP), and stadial (GS-9, 39 ka BP) conditions for the open land LCT. We selected these time slices for their significance in palaeoclimate research. All maps share common features to facilitate their interpretation: Land-cover percentages (REVEALS-based) and pollen percentages are displayed on a gradual colour scale with varying limits to accommodate the vastly different percentages between LCTs and taxa; the differences between REVEALS estimates and pollen counts are also given in percentages on a

diverging colour scale (strong colours denote a strong difference, faded colours denote a small difference, white signals that both values are roughly equal); only sites with data for a given time interval (selected time ± 500 years) are displayed; icon size denotes whether a site is classified as small (<50 ha) or large (>50 ha). Sites marked in red denote unreliable data (see section 4.1 for more details).



## 2.7 R-based application PALVEG

To facilitate the access to our REVEALS land-cover reconstructions of Europe during the Late Pleistocene, we provide an easy-to-use interactive map-generating application: PALVEG, available at https://oakern.shinyapps.io/PALVEG/. PALVEG is a browser-based application using the shiny R package (Chang et al., 2023) and thus does not require any prior coding experience. PALVEG offers easy access to all REVEALS land-cover reconstructions included in this study in the form of dynamically generated maps based on the inputs chosen. PALVEG will be continuously updated with new REVEALS land-

cover estimates as more pollen datasets and REVEALS parameters (e.g., RPP and FSP for additional taxa) become available. PALVEG allows users to define a time slice between 75 and 15 ka BP, in intervals of 1000 years, and chose a taxon or LCT they are interested in. PALVEG will then visualise the associated REVEALS land-cover percentages of all available datasets for the defined time on a map of Europe (10°E–60°W; 30°N–70°N). PALVEG can also display the pollen data as raw count-based percentages for the same interval. To highlight the differences between both methods, users also have the option to

display the deviation of the REVEALS model from raw pollen counts: Blue (positive) values indicate that the REVEALS model attributes a higher land-cover estimate than the count-based estimate. Accordingly, red (negative) values suggest that the REVEALS model assumes a lower land-cover percentage compared to the pollen data. Lastly, PALVEG offers to display both REVEALS land-cover estimates and pollen data aggregated into 1000-year time bins (default) as well as data based on the sample closest to the chosen time slice (within ± 500 years).

We acknowledge the need for intercomparability between different REVEALS land-cover estimate maps, which necessitates a uniform colour scale (i.e., ranging from 0–100 %). Yet, land-cover percentages for many taxa are well below e.g., 10 % across Europe and individual differences are masked by a uniform scale. For maximum clarity, PALVEG therefore offers two modes to display REVEALS land-cover estimates: A fixed scale (0–100 %) and a dynamic scale that changes depending on the maximum land-cover percentages for a given taxon or LCT. This allows for maximum comparability between individual

outputs, while also offering a more detailed and readable display option.

## 3 Results and discussion

In the following, we discuss the spatio-temporal evolution of land-cover occupied by trees in Europe during the Last Glacial using the RPP synthesis described in section 2.3 (Tab. 1). For simplicity, we refer to total tree populations, which are predominantly characterised by conifers, while other LCTs play a minor role. In general, we observe a sharp, Europe-wide

decrease in land-cover of trees starting with the end of MIS 5 (ca. 70 ka BP; Fig. 2). During the subsequent MIS 4 (70–60 ka BP), a minimum in tree cover is observed in all regions with the exception of NE Europe, where the land-cover of total trees remains at a moderate level between 30–40 %, indicating an open woodland vegetation. During early MIS 3 (ca. 60–50 ka BP), tree population recover, particularly in NW and SE Europe in response to the climatic amelioration during prolonged Greenland Interstadials (e.g., GI-16, GI-14, and GI-12). In contrast, tree population decline in NE Europe or are unaffected

(SE Europe) in response to the climatic change. The opposite pattern emerges during later part of MIS 3 (ca. 40–30 ka BP),



when a recovery of tree populations can be observed in NE Europe accompanied by a synchronous, but transient, increase of tree populations in SW Europe. MIS 2 (28–15 ka BP) is generally defined by a subdued variability compared to MIS 3. Land-cover percentages of tree populations in NW and SE Europe are close to their respective minimum for the region, whereas NE and SW Europe signal slightly elevated levels of tree cover.

**Figure 2: Evolution of the mean REVEALS land-cover percentages for all LCTs through time (75–15 ka BP). The data have been categorised in four quadrants split by latitude (> 46° N and < 46° N, ca. North and South of the Alps, respectively) and longitude (> 10.5°E and < 10.5°E, roughly separating the Western Mediterranean from the Central and Eastern Mediterranean)**

Overall, the regional vegetation across Europe is characterised by a division into two major regimes: In NW and SE Europe, the vegetation pattern follows that of other proxy data such as NGRIP $\delta^{18}$O (Fig. 1; Rasmussen et al., 2014): Periods of warmer and more humid conditions coincide with elevated levels of land-cover occupied by trees and vice versa. For NE Europe, and to a lesser extent SW Europe, the opposite pattern emerges. During the coldest and driest intervals (MIS 4 and MIS 2) of the



Last Glacial, tree populations remain at a moderate level and instead decline during the warmer and more humid period of
MIS 3. Although further data is needed to validate this observation, we do not think that the signal from NE Europe is biased
by a combination of generally low pollen productivity and airborne transport of arboreal pollen, mimicking tree stands in the
region. Rather, this see-saw pattern is probably related to large-scale shifts in atmospheric circulation patterns and thus
moisture availability (Florineth and Schlüchter, 2000; Ludwig et al., 2016). Similar observations in speleothems and loess
records have been ascribed to shifts in the North Atlantic Oscillation and associated storm tracks responsible for moisture
distribution (Luetscher et al., 2015) and dust transport (Schaffernicht et al., 2020) across Europe.

In the following, we focus on three critical intervals of the Last Glacial, using the LCT "open land" as proxy for vegetation
openness: Stadial-interstadial variability, represented by GS-9 (39 ka BP) and GI-14 (52 ka BP), respectively, and the Last
Glacial Maximum (LGM, 23 ka BP). To facilitate the comparison, we treat these climatic events as singular time slices centred
around their respective mid-points (given in parentheses). Then, we compare the output of the REVEALS model with pollen
percentages and highlight some of the major differences in the perception of palaeoenvironments. Lastly, we discuss the
reliability of REVEALS estimates in the context of the REVEALS model design and its fundamental ideas.

### 3.1 Stadial-Interstadial variability

For the comparison of interstadial to stadial land cover, we chose the longest interstadial (GI-14) and stadial (GS-9) to fully
capture the differences in land-cover composition. Both climatic events are well-documented in numerous records and
chronologically well-constrained.
In contrast to the interstadial conditions of GI-14, average open land percentages increased to 80 % during Greenland Stadial
9 (GS-9, 39 ka BP) and are highest along the northern fringes of the Alpine glaciers (Fig. 4). This increase is primarily due to
a rise of Poaceae to 32 %, while Cyperaceae (15 %) and Asteraceae (16 %) remained on similar levels as compared to GI-14.
270 Interestingly, Poaceae percentages not only increased during GS-9, but the geographical distribution of Poaceae spread to most
regions of Europe. Notable exceptions are in the Iberian Peninsula, the Alpine Foreland, and the Carpathian Mountains, where
a strong local signal of Cyperaceae indicated the presence of wetlands. Asteraceae and Apiaceae are primarily found in the
Central Mediterranean and South-West Asia, locally reaching more than > 40 % of combined land cover.
Overall, tree populations in Europe collapsed during GS-9. While the western Balkans and Eifel region show a declining
275 presence of tress, locally constrained areas in the Carpathian Mountains (87 % trees) and the Baltics (up to 49 % trees) appear
to have served as refugia for *Picea abies*, *Betula*, and *Pinus* during glacial periods (Fig. 4). However, caution is advised for
the Carpathian data (Fig. 4D). Small, isolated communities of temperate deciduous trees are present in Northern Africa (13
%), the South-West Asia (13 %) and the Eastern Mediterranean (6 %) and comprise the remainder of land covered by trees
during GS-9.

280



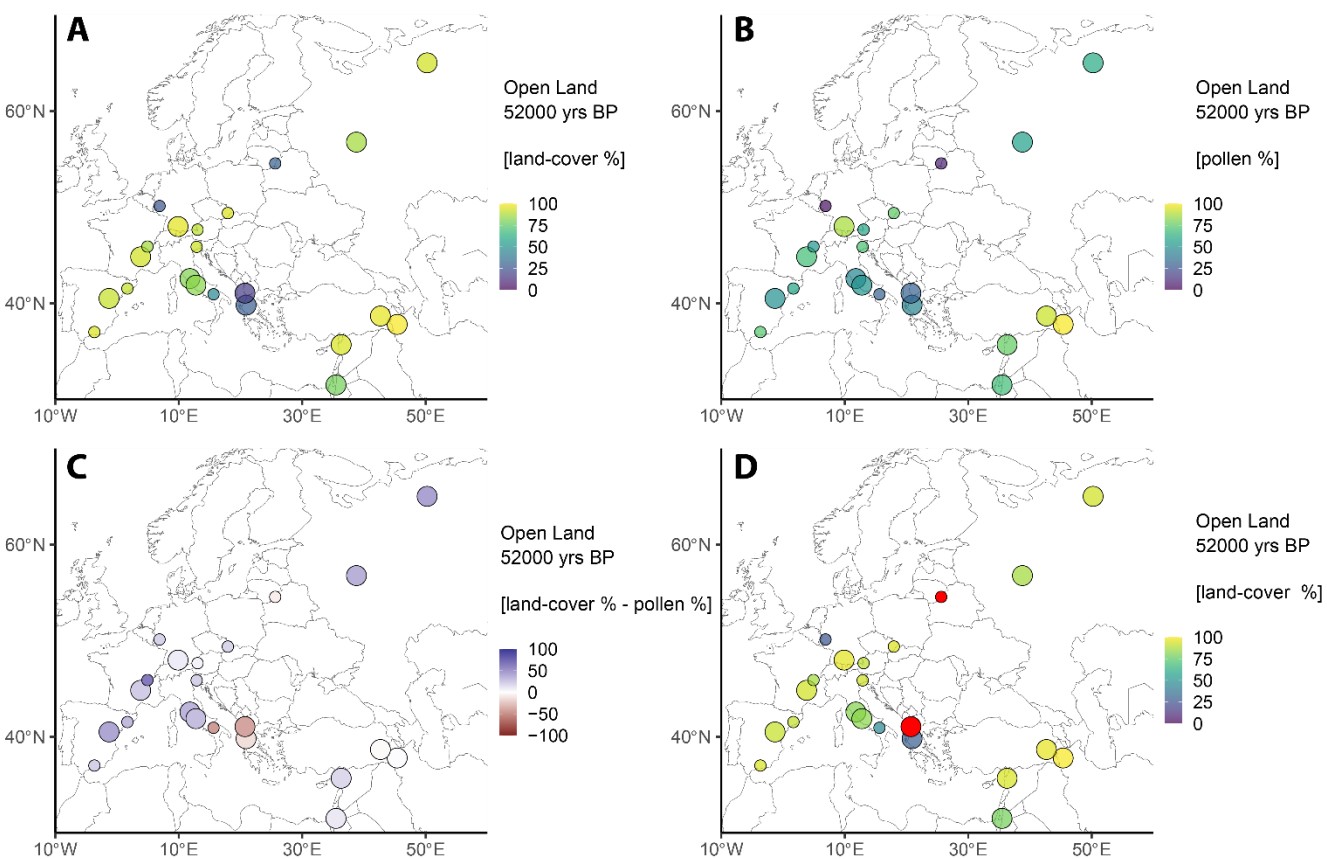

**Figure 3: Results for the LCT "Open land" during GI-14 (52 ka BP). A: REVEALS land-cover percentages. B: Pollen percentages. C: Difference between REVEALS land-cover percentages and pollen data. Colours indicate a higher (blue) or lower (red) land-cover representation compared to pollen data. D: Same as A, but data considered unreliable are highlighted in red (see section 4.1 for details).**

Across all sites, the REVEALS-derived mean land-cover estimate of "open land" during Greenland Interstadial 14 (GI-14, 52 ka BP) was 74 % (Fig. 3). The primary constituents of open land vegetation during GI-14 show distinct geographical patterns: Poaceae (26 %) and Cyperaceae (17 %) are the dominant taxa in Central Europe and the western Mediterranean, whereas Asteraceae (15 %) is found in South-West Asia and the Mediterranean coastal regions. Among other taxa, isolated populations of Apiaceae and Chenopodiaceae are present in South-West Asia and Ranunculaceae reach >10 % in Central European wetlands environments. Other herbs only play subordinate roles.

Although an open steppe-tundra vegetation dominated the European landscapes during GI-14, two distinct forested areas can be identified: The western Balkan and to a lesser extent the Eifel and eastern Baltic regions (Fig. 3A), where conifers constitute the majority of the vegetation. However, caution is advised regarding the reliability of some of these data (Fig. 3D). In contrast, the Italian Peninsula shows a mixed temperate woodland, consisting of deciduous *Quercus*, *Fagus sylvatica*, *Acer*, and *Abies*



*alba* at the time. North of the Alps, forests composed of boreal tree taxa such as *Pinus* and *Betula* constitute a substantial portion of trees while other tree taxa are virtually absent.

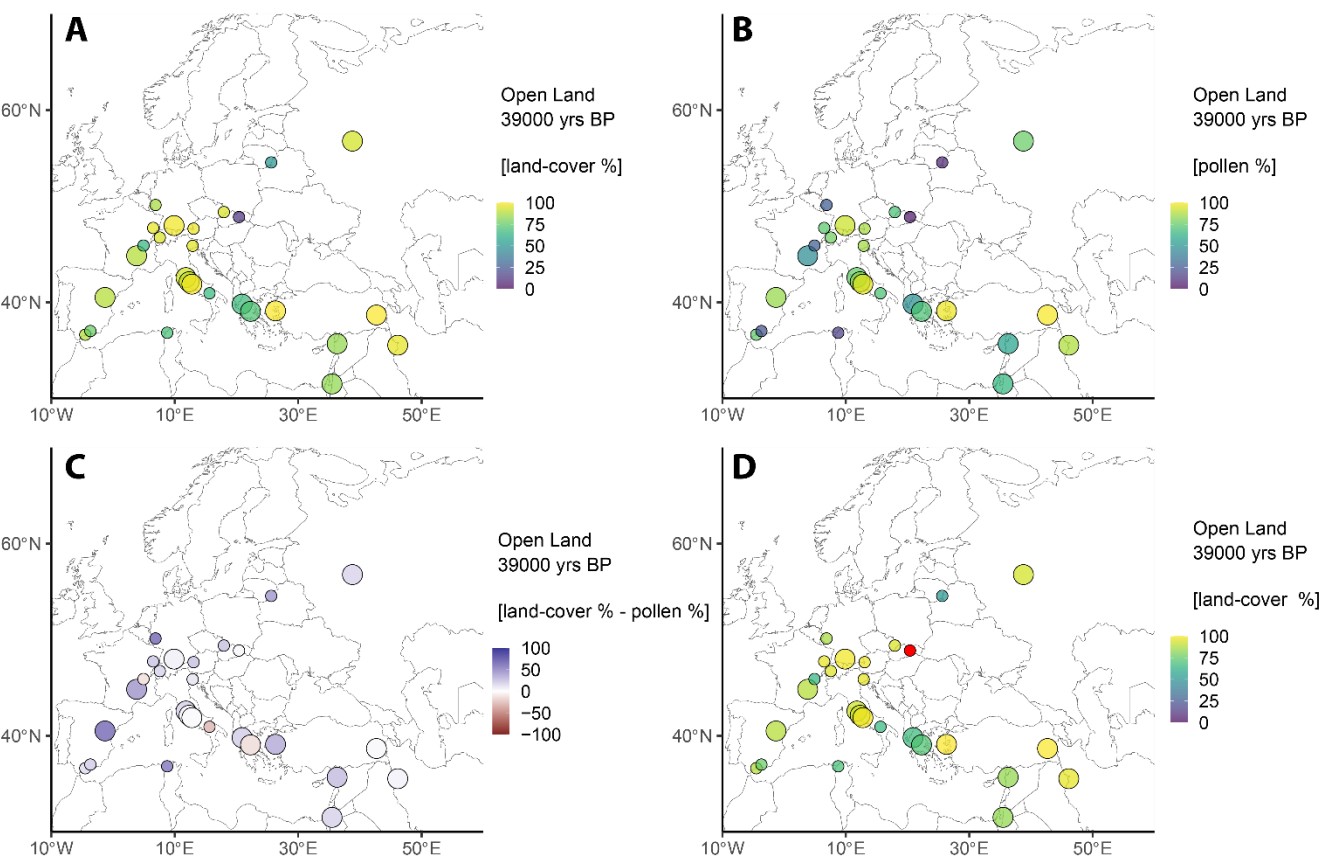

**Figure 4: Results for the LCT "Open land" during GS-9 (39 ka BP). A: REVEALS land-cover percentages. B: Pollen percentages. C: Difference between REVEALS land-cover percentages and pollen data. Colours indicate a higher (blue) or lower (red) land-cover representation compared to pollen data. D: Same as A, but data considered unreliable are highlighted in red (see section 4.1 for details).**

The stadial-interstadial variability is primarily characterised by an increase in open-land percentages, which is in line with the interpretation of proxy records across Europe (Fletcher et al., 2010; Landais et al., 2022). This increase is accompanied by northward propagation of open land across Central Europe. The scarcity of available pollen records for GI-9 from Northern and North-eastern Europe substantially hinders our ability to further inquire, but existing data suggest that a tree-line shift to higher latitudes might have occurred. However, both areas are geographically limited by the extent of the Fennoscandian ice sheet (Fig. 1). Additional pollen records are required to further investigate the spatio-temporal framework of tree-line recession in a north-easterly direction.

## 3.2 Last Glacial Maximum

REVEALS land-cover estimates for the LGM are characterised by uniformly high percentages of open land vegetation (88 %) across most of Europe. The decrease in forested area by 40 % from GS-9 (or 48 % from GI-14) demonstrates the substantial impact of the increasingly cool and dry climates on the European vegetation going from MIS 3 to MIS 2. Similar to the previous
intervals, the main constituents are Poaceae (31 %), Cyperaceae (18 %), and Asteraceae (23 %), although their geographic distribution varies substantially. Poaceae is mainly found in southern Central Europe and the Western Mediterranean region, while Cyperaceae is primarily found at higher latitudes (> 46 ° N) and Asteraceae at lower latitudes (< 46 °N). Similar to previously discussed intervals, Apiaceae is most prominent in South-West Asia (>30 %). Substantial amounts of *Artemisia* (up to 37 %) and Chenopodiaceae (up to 10 %) are scattered across Europe.


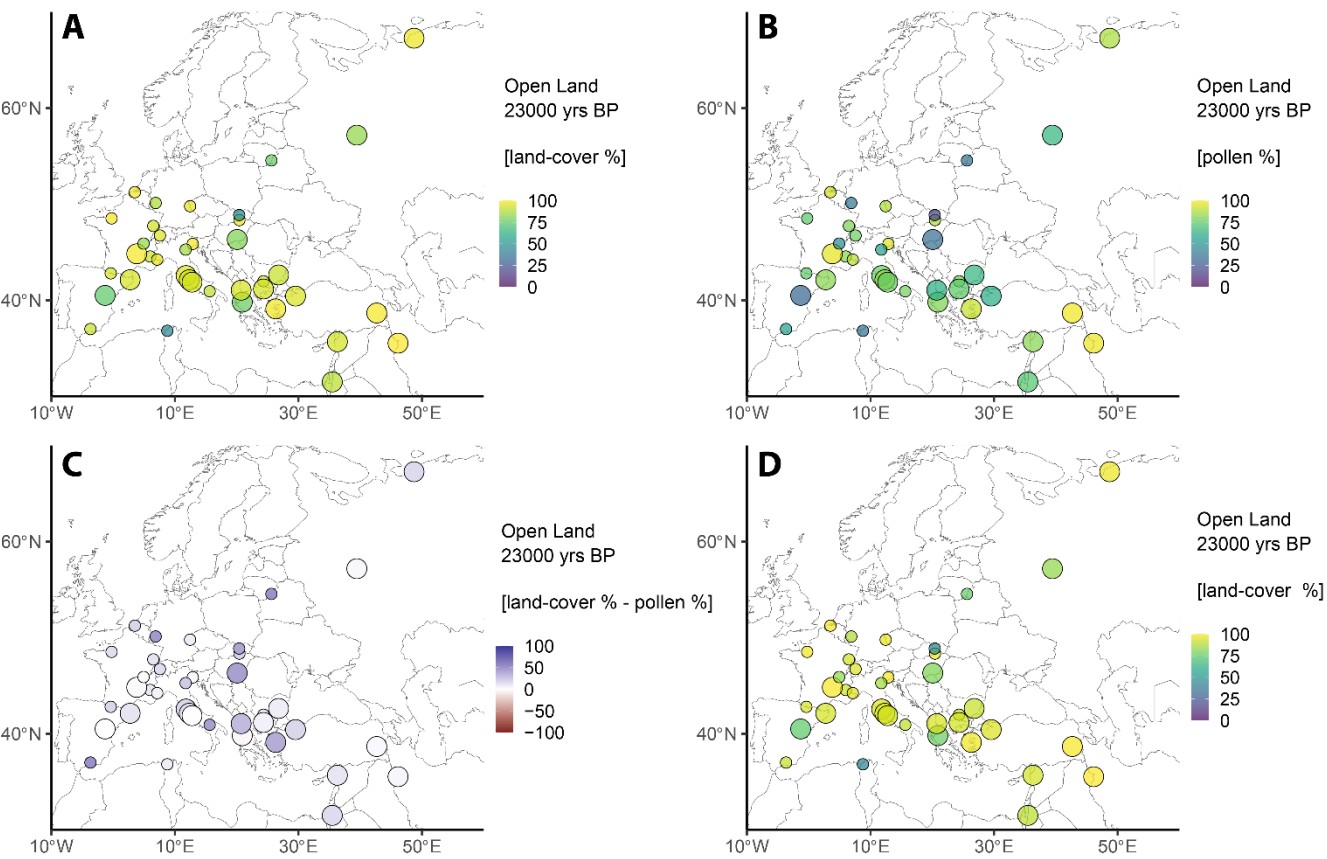

**Figure 5: Results for the LCT "Open land" during the LGM (23 ka BP). A: REVEALS land-cover percentages. B: Pollen percentages. C: Difference between REVEALS land-cover percentages and pollen data. Colours indicate a higher (blue) or lower (red) land-cover representation compared to pollen data. D: Same as A, but data considered**
**unreliable are highlighted in red (see section 4.1 for details).**

Although tree populations are close to their minimum extent during the LGM (Fig. 5A), two locally constrained glacial refugia exist, where trees constitute more than 30 % land cover: In the Carpathian Mountains, *Picea abies* and *Pinus* demarcate an open coniferous woodland vegetation. Similarly, strands of boreal forests composed of *Picea abies*, *Pinus*, and *Betula* prevail in the Balkans. This presence of substantial tree populations highlights the existence of glacial tree refugia with suitable

microclimates during the LGM.

Among other sites, the total tree-cover during the LGM is very low and reaches a maximum of 25 %. Nonetheless, geographical patterns are discernible from the REVEALS estimates. Small populations of *Pinus* are present in a punctuated, SW-NE belt spanning from Iberia to Russia, while being absent in the Central and Western Mediterranean as well as North-western Europe. Cold deciduous trees and temperate deciduous trees are scattered across Europe but only occur locally in low percentages

(typically < 10 %). Noticeable amounts of Mediterranean tree taxa during the LGM are only attested in Greece (*Carpinus orientalis*; 1.5 %).

**3.3 Comparison of REVEALS estimates and pollen data**

The direct comparison between REVEALS land-cover estimates and raw pollen percentages highlights the differences between both approaches (Figs. 3C, 4C, and 5C). Expectedly, an overrepresentation of trees in pollen records (and consequently much

lower REVEALS land-cover estimates) can be observed (Fig. 6). Accordingly, grasses and herbs are underrepresented in pollen data and achieve much higher REVEALS land-cover estimates. However, on a taxon level, this overarching pattern may not always apply and individual taxon-specific land-cover percentages ultimately depend on the overall vegetational composition. REVEALS estimates may be lower or higher for certain taxa than the respective pollen percentages (e.g., for Poaceae and Cyperaceae). In most cases, this is associated with the taxon-specific RPP values in relation to the overall RPP

values of all other taxa present at the site (Tab. 1). Notable outlier to this pattern is *Abies alba*, which despite relatively high RPP values is considered underrepresented by the REVEALS model, owing to the high fall speed (FSP) of *Abies alba* pollen grains and therefore its limited spatial dispersal capabilities.

As a result of the disparity between REVEALS data and fossil pollen data, the presence of forests and (open) woodlands may be exaggerated when interpreting fossil pollen data. However, the extent of e.g., the over-representation of tree taxa is highly

variable through time and land-cover estimates require careful analysis in order to fully utilise their potential. Overall, these observations are in line with the general conception among palynologists and palaeobotanists that fossil pollen data do not reflect the palaeovegetation one-to-one, which ultimately led to the inception of models such as REVEALS (Andersen, 1970; Prentice, 1985; Sugita, 2007).

**3.4 The impact of RPP on REVEALS land-cover estimates**

In section 2.3 we have discussed the existence of various compilations and syntheses of RPP and FSP values. While our study used a synthesis of RPP data from Githumbi et al. (2022) and Serge et al. (2023), we want to briefly address the differences in REVEALS land-cover estimates using other sets of RPP values (Tab. 1). All available datasets are established using high-



quality estimates of pollen productivity compiled from research of the last decades and all RPP datasets focus on slightly different aspects in terms of scale regional (e.g., Mediterranean) to global, environmental variability, and methodology. Hence,

there is no *best* dataset for all purposes.

In general, all RPP datasets yield a vegetation composition that indicates an overrepresentation of trees in fossil pollen data and consequently much higher land-cover estimates for "open land" (Fig. 6). However, there are several distinct differences between the RPP datasets. First and foremost, the results for the synthesis used in this study and the compilation by Wieczorek and Herzschuh (2020) are strikingly similar, with the latter dataset generating slightly higher land-cover estimates for the

different tree LCTs. In contrast, land-cover estimates using the optimized RPP (Schild et al., 2024) result in very low percentages of tree LCTs. In detail, this primarily affects "conifers", influenced by the vastly higher RPP of *Pinus* and to a lesser extent *Picea abies*, which both affect land-cover estimates negatively. Other tree LCTs are in good agreement with those of other RPP datasets. Differences among herbaceous taxa vary across RPP datasets, but are less significant on regional scales.

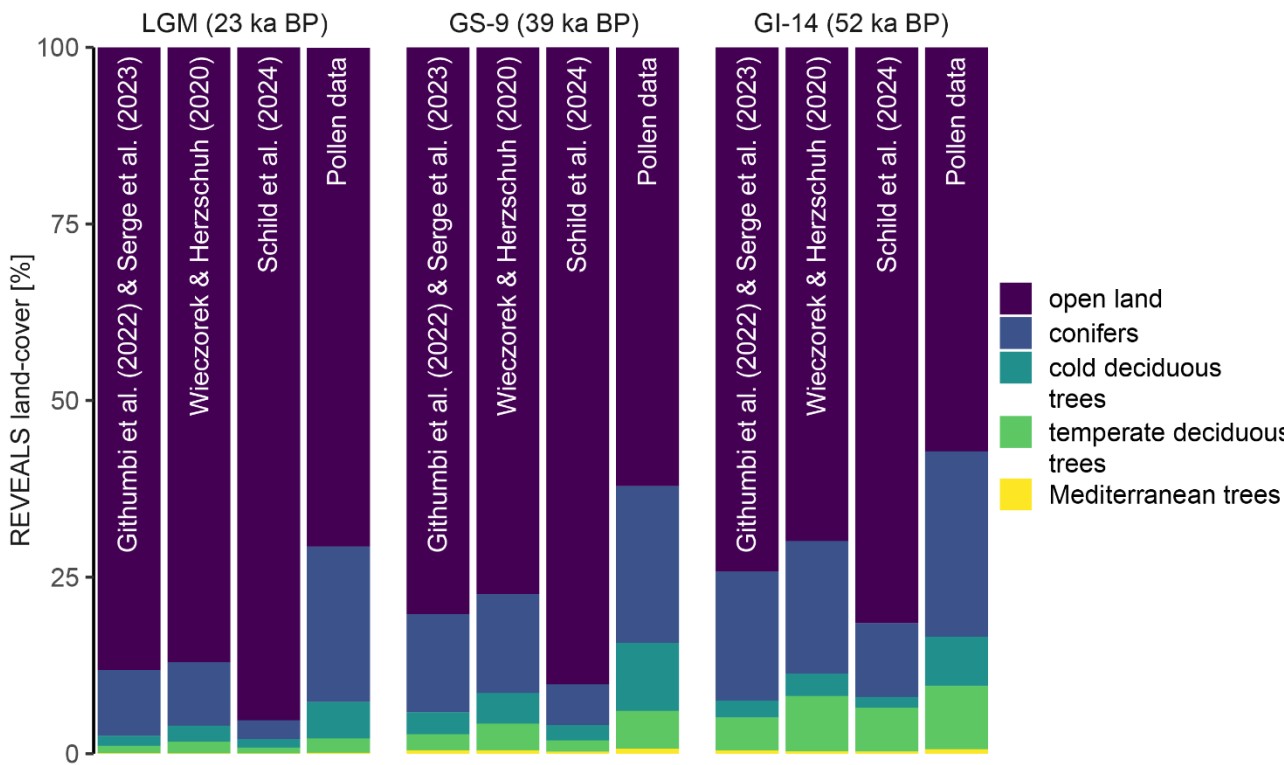


**Figure 6: Comparison of REVEALS land-cover percentages for three critical time intervals, using different datasets of relative pollen productivity (RPP) as discussed in section 2.3 (Tab. 1). The data are compared to the fossil pollen percentages.**



## 4 Data reliability

The reliability of REVEALS land-cover estimates is derived from the standard errors (SEs) that are provided alongside each
REVEALS model run (Serge et al., 2023; Trondman et al., 2015). The relative size of SEs is considered a measure of the
quality of the REVEALS output. REVEALS land-cover SEs mainly rely on the quantity and quality of the available pollen
datasets. This includes both the number of available datasets as well as their individual counting sums for each sample and the
classification into large (>50 ha) and small (<50 ha) lakes and bogs (Githumbi et al., 2022; Sugita, 2007). Additionally, the
standard deviation of the REVEALS parameter RPP is factored into the estimation of the REVEALS SEs.

A key aspect of previous land-cover reconstructions using the REVEALS model for the Holocene period is to present the
results in 1° x 1° grid cells, which aggregate data from all sites within a grid cell, based on the type and size of a basin (e.g.,
Githumbi et al., 2022; Serge et al., 2023; Trondman et al., 2016). The idea is that the quality of REVEALS estimates for a grid
cell increases with the number of large lakes present within each grid cell (Sugita, 2007). Later studies have demonstrated that
pollen data from small sites (<50 ha) will also improve REVEALS estimates, although to a lesser extent (Fyfe et al., 2013;
Trondman et al., 2016). Hence, the quality of vegetation estimates has been related to the number and basin size of the archives
where pollen records originate from. For the Last Glacial period, this approach is not feasible due to the low geographical
cover of suitable (i.e., chronologically well-constrained) pollen archives in Europe (Fig. 1). Based on the current data
compilation, only three 1° x 1° grid cells contain more than one pollen archive from the Last Glacial period. Therefore, we
provide point-based land-cover estimates that represent the local to regional vegetation within a maximum radius of 100 km
around the site (as defined in the REVEALS parameter Zmax).

### 4.1 Estimation of uncertainty

To evaluate whether REVEALS estimates are reliable, we follow the protocol of Githumbi et al. (2022): If the standard
deviation of the REVEALS estimate is equal to or larger than the REVEALS estimate itself, then the data is considered
unreliable. This excludes entries with zero counts (and thus a REVEALS land-cover of 0 %), which are considered reliable by
definition. In total, 39 % of all non-zero REVEALS land-cover estimates are considered unreliable (Tab. D1). However, a
substantial amount of unreliable data is related to taxa that only appear in very low counts (e.g., *Urtica*, Rubiaceae, *Potentilla*,
and *Filipendula*), which are expected to have a high standard error (Sugita, 2007) and/or have high RPP standard deviations
(e.g., *Abies alba* and *Populus* for which no data is considered reliable). In contrast, taxa that are commonly found in glacial
climates such as Poaceae (2 %), Cyperaceae (8 %) or *Pinus* (2 %) are on average considered highly reliable. Establishing LCTs
further increases the reliability compared to the individual reliability of most their constituents, e.g., only 2 % of open land
and 14 % of total trees datapoints considered unreliable.

Due to the fact that many taxa are absent in a lot of the samples, the perceived reliability of REVEALS land-cover estimates
is much higher. For example, only 8 % of all *Urtica* samples are considered unreliable if all samples are considered. However,
95 % of datapoints are classified as unreliable if zero-count samples are excluded. Hence, we advise caution regarding the use

and interpretation of data from relatively uncommon taxa. Instead, LCTs represent a broader spectrum of the vegetation and
their data are more reliable and thus less prone to misinterpretation.

## 4.2 Data aggregation

Previous studies have suggested to aggregate samples from intervals with a high temporal density in order to increase the size
of pollen counts and thus minimise REVEALS SEs (Githumbi et al., 2022; Trondman et al., 2016). In this study, samples
± 500 years around the studied time interval were aggregated into a single sample, which was then used to estimate REVEALS
land-cover percentages. As a result, the input dataset shrunk from 5231 to 1632 samples, corresponding to an average
aggregation ratio of 3.21 to 1 across all samples. However, due to the unevenly temporally spaced nature of palynological
datasets, the number of aggregated samples per bin may vary substantially. The number of sites that contain data for each time
step are given in Appendix B (Fig. B2).

Overall, data aggregation decreased the number of unreliable REVEALS land-cover estimates from 48 % (raw pollen counts)
to 40 % (aggregated pollen counts). The percentage of unreliable data was reduced drastically for almost all taxa and LCTSs.
The biggest improvements occur for taxa with low to very low counting sums per sample, such as *Carpinus orientalis* (39 %
more reliable), *Buxus sempervirens* (20 % more reliable), *Acer* (18 % more reliable). Exceptions are Poaceae (0.3 % less
reliable) and *Abies alba* (no improvement). Among LCTs, the reliability of estimates for cold deciduous trees, temperate
deciduous trees and Mediterranean trees increases (9 %, 8 %, and 11 %, respectively), whereas the reliability of estimates for
conifers decreases slightly (0.3 %). In conclusion, data aggregation has substantially improved to REVEALS land-cover
estimates among less common taxa and LCTs, whereas estimates for the dominant taxa and LCTs remain virtually unchanged.
The percentages of unreliable REVEALS land-cover estimates for both aggregated and raw data can be found in the
Appendix D (Tab. D1). Based on our findings, we suggest to use the REVEALS land-cover estimates from aggregated data
due to the predominantly higher data reliability observed. Nonetheless, to provide maximum clarity, we provide REVEALS
land-cover estimates for both the aggregated and raw datasets and also include both datasets in the PALVEG application.

## 5 Code availability

The REVEALS function is part of the R package LRA (Abraham et al., 2014), available at https://github.com/petrkunes/LRA
(last accessed: 29 April 2024). The full code used in this study is available on Zenodo (DOI: 10.5281/zenodo.12625221) (Kern,
2024b). It includes data preparation, taxon harmonization, and data analysis.
The application PALVEG is hosted at https://oakern.shinyapps.io/PALVEG/ and the R code is available on Zenodo (DOI:
10.5281/zenodo.12624842) (Kern, 2024a).



## 6 Data availability

REVEALS land-cover estimates and REVEALS standard errors generated in this study are available in the PANGAEA
database (www.pangaea.de): https://www.pangaea.de/tok/894b8daa21b7143a77e5cd9f1b32f23f7b73b42d (Dataset in review;
temporary access for reviewers, final DOI will be generated upon reviewers approval)

## 7 Conclusions

Over the recent years, the REVEALS model has been successfully implemented in numerous research projects, such as
LandClim (e.g., Githumbi et al., 2022) and PAGES LandCover6k (e.g., Harrison et al., 2020), as well as studies (e.g., Roberts
et al., 2018; Kaplan et al., 2017) in the past. The results have been utilised in a wide array of research disciplines including but
not limited to anthropogenic land-cover change, to evaluate archaeology-based reconstruction of land-use in the past (Morrison
et al., 2021), to estimate the impact of mining activities (Schauer et al., 2019), or in the context of biodiversity and regional to
local scale vegetation dynamics (Marquer et al., 2017).

For the first time, the REVEALS model has been applied to pollen records from Europe during the latest interval of the Last
Glacial period, generating continuous (1000-year intervals) land-cover estimates for 38 taxa and 5 LCTs. The Last Glacial is
of particular importance for our understanding of abrupt climate change (i.e., GI/GS variability) and the response of ecosystems
(e.g., Fletcher et al., 2010; Landais et al., 2022; Rasmussen et al., 2014). Yet, to this day, vegetation dynamics in Europe during
the Last Glacial are far from understood and current hypotheses include rapid re-migration of tree taxa in response to climate
amelioration, local refugia providing suitable microclimates that sheltered from the harsh environmental conditions, and
complex spatio-temporal combinations of the aforementioned mechanisms (Gavin et al., 2014; Hošek et al., 2024; Tzedakis et
al., 2013). Moreover, the Last Glacial saw the disappearance of Neanderthals and the dispersal of *Homo sapiens* into Europe
followed by pronounced demographic fluctuations (Hublin, 2015; Mellars, 2004; Schmidt et al., 2021). It has long been
hypothesised to what extent climatic fluctuations have impacted these patterns of population dynamics on both temporal and
spatial scales (Maier et al., 2022; Staubwasser et al., 2018; Timmermann and Friedrich, 2016). More sophisticated vegetation
reconstructions might shine light on open research questions regarding food and prey availability, hunting and subsistence
strategies, and demographic developments (Baker et al., 2024; Ordonez and Riede, 2022; Schmidt and Zimmermann, 2019;
Seersholm et al., 2020; Vidal-Cordasco et al., 2023).

With the PALVEG application, we aim to facilitate the access to palaeovegetation reconstructions to a wide scientific audience.
PALVEG dynamically generates maps of REVEALS land-cover estimate or pollen data for all sites, taxa, LCTs, and time
intervals between 75 and 15 ka BP. PALVEG is browser-based, intuitive, and requires no *a priori* experience in coding or
palaeobotany.



## 8 Outlook

Further improving the REVEALS model highly depends on the availability and quality of plant physiological parameters (RPP and FSP). Researching pollen productivity and dispersal is tedious and relies on numerous field studies and although steady

progress has been made over the last couple of years, more RPP and FSP data are required. This is of particular significance for taxa that are presumed to shape the European landscape during glacial times (e.g., differentiating between tree and shrub growth forms for taxa such as *Betula* and *Pinus*; Birks, 1968) or those that are well-known for their proportionately high or low pollen productivity (e.g., *Salix* and *Larix*; Niemeyer et al., 2015) and are thus misrepresented in pollen records. Hence, the most promising avenue to further improve land-cover estimates is to provide these parameters for more taxa, ideally on the

species or genus level, to fully utilise the information that available pollen datasets provide. Specifically for the Late Pleistocene in Europe, more pollen data at a sufficiently high temporal resolution are required to further refine the geographical cover of available data and the inferred palaeoecological reconstructions. Ideally, these new data originate from large lakes (>50 ha) and fill geographical gaps in areas of Europe where the present distance between available archives is particularly large, e.g., Central and Eastern Europe (Fig. 1). A different approach to improve REVEALS-based palaeovegetation models

lies optimisation of RPP using remote-sensing techniques, which has already shown promising results and is expect to further improve in the future.

In the face of anthropogenic climate change, generating high-quality palaeovegetation data remains imperative for improving coupled vegetation-climate models. Only then can ESMs provide more accurate predictions of vegetation and ecosystem dynamics in the near future.




## Appendix A: Pollen archives used in this study

**Table A1: List of all pollen archives, REVEALS land-cover reconstructions and their respective references. "Site ID" refers to the labels in Fig. 1.**

| Site ID | Site | Database | Reference |
|---|---|---|---|
| 1 | Timan Ridge | Neotoma | Paus, A., Svendsen, J. I., & Matiouchkov, A. (2003). Late weichselian (valdaian) and Holocene vegetation and environmental history of the northern Timan Ridge, European arctic Russia. Quaternary Science Reviews, 22(21-22), 2285-2302. |
| 2 | Lake Yamozero | Neotoma | Henriksen, M., Mangerud, J. A. N., Matiouchkov, A., Murray, A. S., Paus, A., & Svendsen, J. I. (2008). Intriguing climatic shifts in a 90 kyr old lake record from northern Russia. Boreas, 37(1), 20-37. |
| 3 | Lake Nero | Neotoma | Sudakova, N.G., V.V. Dashev, and V.V. Pisareva. 1984. Field guide book for Excursion 10-B. 27th International Geological Congress Moscow. Russia. |
| 4 | Plesheevo Lake | Neotoma | Wohlfarth, B., Tarasov, P., Bennike, O., Lacourse, T., Subetto, D., Torssander, P., & Romanenko, F. (2006). Late glacial and Holocene palaeoenvironmental changes in the Rostov-Yaroslavl'area, West Central Russia. Journal of Paleolimnology, 35, 543-569. |
| 5 | Medininkai | - | Šeirienė, V., Kühl, N., & Kisielienė, D. (2013). Quantitative reconstruction of Eemian (Merkine) and Weichselian (Numunas) climate in Lithuania. Palaeolandscapes from Saalian to Weichselian, 92. |
| 6 | Glendalough Valley | - | Mitchell, F.J.G., Maldonado-Ruiz, J., 2018. Vegetation development in the Glendalough Valley, eastern Ireland over the last 15,000 years. Biology and Environment: Proceedings of the Royal Irish Academy 118B, 55–68. |
| 7 | Reenadinna Wood | Neotome | Mitchell, F.J.G., 1990. The history and vegetation dynamics of a yew wood (Taxus baccata L.) in S.W. Ireland. New Phytologist 115, 573–577. |
| 8 | Terneuzen | Neotoma | Verbruggen, F., van der Linden, M., Kooistra, L. I., & Beurden, L. (2015). Stille wateren hebben diepe gronden: paleoecologisch onderzoek aan het Pleistoceen en Holoceen van Terneuzen, Nieuwe Sluis. BIAX Consult. |
| 9 | Moershoofd | Neotoma | Zagwijn, W. H. (1974). Vegetation, climate and radiocarbon datings in the Late Pleistocene of the Netherlands. II. Middle Weichselian. Meded. Rijks Geol. Dienst, N.S., 14(2) & 25(3). |
| 10 | La Grande Pile | - | De Beaulieu, J. L., & Reille, M. (1992). The last climatic cycle at La Grande Pile (Vosges, France) a new pollen profile. Quaternary Science Reviews, 11(4), 431-438. |
| 11 | ELSA | - | Britzius, S., Dreher, F., Maisel, P., & Sirocko, F. (2024). Vegetation Patterns during the Last 132,000 Years: A Synthesis from Twelve Eifel Maar Sediment Cores (Germany): The ELSA-23-Pollen-Stack. Quaternary, 7(1), 8. |
| 12 | Brentenlohe | Neotoma | Knipping, M. (1997). Pollenanalytische Untersuchungen zur Siedlungsgeschichte des Oberpfälzer Waldes. TELMA-Berichte der Deutschen Gesellschaft für Moor-und Torfkunde, 27, 61-74. |
| 13 | Jammertal | Neotoma | Müller, U. C. (2000). A Late-Pleistocene pollen sequence from the Jammertal, south-western Germany with particular reference to location and altitude as factors determining Eemian forest composition. Vegetation History and Archaeobotany, 9, 125-131. |
| 14 | Safarka | PalyCZ | Jankovská, V. (2008). Slovak and Moravian Carpathians in the last glacial period–an island of "Siberian taiga" in Europe. Phytopedon, 7, 122-130. |
| 15 | Saint Ursin | Neotoma | Barbier, D. (1999). Histoire de la végétation du nord-mayennais de la fin du Weichsélien à l'aube du XXIe siècle. Mise en évidence d'un Tardiglaciaire armoricain. Interactions Homme-Milieu. |
| 16 | Nagy-Mohos-to | Neotoma | Magyari, E. K., Kuneš, P., Jakab, G., Sümegi, P., Pelánková, B., Schäbitz, F., ... & Chytrý, M. (2014). Late Pleniglacial vegetation in eastern-central Europe: are there modern analogues in Siberia?. Quaternary Science Reviews, 95, 60-79. |
| 17 | Kopais | Neotoma | Turner, J., & Greig, J. R. (1975). Some Holocene pollen diagrams from Greece. Review of Palaeobotany and Palynology, 20(3), 171-204. |
| 18 | Le Fourneau | Neotoma | Barbier, D., Visset, L., 2000. Les spécificités d'un Tardiglaciaire armoricain : étude pollinique synthétique à partir de trois tourbières du nord-est mayennais (France) [Specificities of an armoricain Late Glacial period : synthetic pollen analysis of three peat bogs in northeast Mayenne (France)]. Quaternaire 11, 99–106. |
| 19 | Füramoos | - | Kern, O. A., Koutsodendris, A., Allstädt, F. J., Mächtle, B., Peteet, D. M., Kalaitzidis, S., ... & Pross, J. (2022). A near-continuous record of climate and ecosystem variability in Central Europe during the past 130 kyrs (Marine Isotope Stages 5–1) from Füramoos, southern Germany. Quaternary Science Reviews, 284, 107505. |



| 20 | Ioannina | Neotoma | Tzedakis, P. C., Frogley, M. R., Lawson, I. T., Preece, R. C., Cacho, I., & De Abreu, L. (2004). Ecological thresholds and patterns of millennial-scale climate variability: The response of vegetation in Greece during the last glacial period. Geology, 32(2), 109-112. |
|----|----------|---------|------|
| 21 | Nesseltalgraben | - | Mayr, C., Stojakowits, P., Lempe, B., Blaauw, M., Diersche, V., Grohganz, M., Correa, M.L., Ohlendorf, C., Reimer, P., Zolitschka, B., 2019. High-resolution geochemical record of environmental changes during MIS 3 from the northern Alps (Nesseltalgraben, Germany). Quaternary Science Reviews 218, 122–136. |
| 22 | Lobsigensee | Neotoma | Ammann, B. (1989). Late-quaternary palynology at Lobsigensee. Regional vegetation history and local lake development. Dissertationes Botanicae, 137. |
| 23 | Amsoldingersee | Neotoma | Lotter, A., & Boucherle, M. M. (1984). A late-glacial and post-glacial history of Amsoldingersee and vicinity, Switzerland. Schweizerische Zeitschrift für Hydrologie, 46, 192-209. |
| 24 | Feher Lake | Neotoma | Magyari, E. K., Kuneš, P., Jakab, G., Sümegi, P., Pelánková, B., Schäbitz, F., ... & Chytrý, M. (2014). Late Pleniglacial vegetation in eastern-central Europe: are there modern analogues in Siberia?. Quaternary Science Reviews, 95, 60-79. |
| 25 | Les Echets | ACER | de Beaulieu, Jacques-Louis; Reille, Maurice (1984): A long Upper Pleistocene pollen record from Les Echets, near Lyon, France. Boreas, 13(2), 111-132. |
| 26 | Azzano Decimo | ACER | Pini, Roberta; Ravazzi, Cesare; Donegana, D (2009): Pollen stratigraphy, vegetation and climate history of the last 215ka in the Azzano Decimo core (plain of Friuli, north-eastern Italy). Quaternary Science Reviews, 28(13-14), 1268-1290. |
| 27 | Lago della Costa | Neotoma | Kaltenrieder, P., Procacci, G., Vannière, B., Tinner, W., 2010. Vegetation and fire history of the Euganean Hills (Colli Euganei) as recorded by Lateglacial and Holocene sedimentary series from Lago della Costa (northeastern Italy). The Holocene 20, 679–695. |
| 28 | Lac du Bouchet | ACER | de Beaulieu, J. L., & Reille, M. (1992). Long Pleistocene pollen sequences from the Velay Plateau (Massif Central, France) I. Ribains maar. Vegetation History and Archaeobotany, 1, 233-242. |
| 29 | Brameloup | Neotoma | De Beaulieu, J.L., Pons, A., Reille, M., 1985. Recherches pollenanalytiques sur l'histoire tardiglaciaire et holocene de la vegetation des Monts d'Aubrac (Massif Central, France). Review of Palaeobotany and Palynology 44, 37–80. |
| 30 | Lac de Siguret | Neotoma | de Beaulieu, J.L., 1977. Contribution pollenanalytique à l'histoire tardiglaciaire et holocène de la végétation des Alpes méridionales françaises (PhD Thesis). éditeur non identifié. |
| 31 | Correo | Neotoma | Nakagawa, T. (1998). Études palynologiques dans les Alpes françaises centrales et méridionales: histoire de la végétation tardiglaciaire et holocène (Doctoral dissertation, Aix-Marseille 3). |
| 32 | Laghi dell'Orgials | Neotoma | Ortu, E., Brewer, S., & Peyron, O. (2006). Pollen-inferred palaeoclimate reconstructions in mountain areas: problems and perspectives. Journal of Quaternary Science: Published for the Quaternary Research Association, 21(6), 615-627. |
| 33 | Verdeospesoa mire | Neotoma | Pérez-Díaz, S., López-Sáez, J.A., 2017. 33. Verdeospesoa mire (Basque Country, Northern Iberian Peninsula, Spain). Grana 56, 315–317. |
| 34 | Lago de Ajo | Neotoma | Allen, J.R.M., Huntley, B., Watts, W.A., 1996. The vegetation and climate of northwest Iberia over the last 14,000 years. Journal of Quaternary Science 11, 125–147. |
| 35 | El Portalet | - | González-Sampériz, P., Valero-Garcés, B. L., Moreno, A., Jalut, G., García-Ruiz, J. M., Martí-Bono, C., ... & Dedoubat, J. J. (2006). Climate variability in the Spanish Pyrenees during the last 30,000 yr revealed by the El Portalet sequence. Quaternary Research, 66(1), 38-52. |
| 36 | Straldzha mire | Neotoma | Tonkov, S., Lazarova, M., Bozilova, E., Ivanov, D., & Snowball, I. (2014). A 30,000-year pollen record from Mire Kupena, Western Rhodopes Mountains (south Bulgaria). Review of palaeobotany and palynology, 209, 41-51. |
| 37 | Lagaccione | ACER | Magri, Donatella (1999): Late Quaternary vegetation history at Lagaccione near Lago di Bolsena (central Italy). Review of Palaeobotany and Palynology, 106(3-4), 171-208. |
| 38 | Lake Banyoles | ACER | Pérez-Obiol, Ramon P; Julia, R (1994): Climatic change on the Iberian Peninsula recorded in a 30,000-year pollen record from Lake Banyoles. Quaternary Research, 41(1), 91-98. |
| 39 | Straciacappa | ACER | Giardini, Marco (2006): Late Quaternary vegetation history at Stracciacappa (Rome, central Italy). Vegetation History and Archaeobotany, 16(4), 301-316. |
| 40 | Labsky Dul | Neotoma | Engel, Z., Nývlt, D., Křížek, M., Treml, V., Jankovská, V., & Lisá, L. (2010). Sedimentary evidence of landscape and climate history since the end of MIS 3 in the Krkonoše Mountains, Czech Republic. Quaternary Science Reviews, 29(7-8), 913-927. |
| 41 | Valle di Castiglione | ACER | Alessio, Marisa; Allegri, Lucia; Bella, Francesco; Calderoni, Gilberto; Cortesi, Cesarina; Dai Pra, Giuseppe; De Rita, Donatella; Esu, Daniela; Follieri, Maria; Improta, Salvatore; Magri, Donatella; Narcisi, Biancamaria; Petrone, Vincenzo; Sadori, Laura (1986): 14C dating, |





| | | | |
|---|---|---|---|
| | | | geochemical features, faunistic and pollen analyses of the uppermost 10 m core from Valle di Castiglione (Rome, Italy). Geologica Romana, 25. |
| 42 | Abric Romani | ACER | Burjachs, F., & Julià, R. (1994). Abrupt climatic changes during the last glaciation based on pollen analysis of the Abric Romani, Catalonia, Spain. Quaternary Research, 42(3), 308-315. |
| 43 | Tenaghi Philippon | - | Koutsodendris, A., Dakos, V., Fletcher, W. J., Knipping, M., Kotthoff, U., Milner, A. M., ... & Pross, J. (2023). Atmospheric CO2 forcing on Mediterranean biomes during the past 500 kyrs. Nature Communications, 14(1), 1664. |
| 44 | Lake Ohrid | - | Sadori, L., Koutsodendris, A., Panagiotopoulos, K., Masi, A., Bertini, A., Combourieu-Nebout, N., ... & Donders, T. H. (2016). Pollen-based paleoenvironmental and paleoclimatic change at Lake Ohrid (south-eastern Europe) during the past 500 ka. Biogeosciences, 13(5), 1423-1437. |
| 45 | Lago Grande di Monticchio | ACER | Brauer, Achim; Allen, Judy R M; Mingram, Jens; Dulski, Peter; Wulf, Sabine; Huntley, Brian (2007): Evidence for last interglacial chronology and environmental change from Southern Europe. Proceedings of the National Academy of Sciences, 104(2), 450-455. |
| 46 | Villarquemado | - | González-Sampériz, P., Gil-Romera, G., García-Prieto, E., Aranbarri, J., Moreno, A., Morellón, M., ... & Valero-Garcés, B. L. (2020). Strong continentality and effective moisture drove unforeseen vegetation dynamics since the last interglacial at inland Mediterranean areas: The Villarquemado sequence in NE Iberia. Quaternary Science Reviews, 242, 106425. |
| 47 | Lake Iznik | Neotoma | Miebach, A., Niestrath, P., Roeser, P., & Litt, T. (2016). Impacts of climate and humans on the vegetation in northwestern Turkey: palynological insights from Lake Iznik since the Last Glacial. Climate of the Past, 12(2), 575-593. |
| 48 | Jablunka | PalyCZ | Jankovská, V. (2008). Slovak and Moravian Carpathians in the last glacial period–an island of "Siberian taiga" in Europe. Phytopedon, 7, 122-130. |
| 49 | Megali Limni | ACER | Margari, V., Gibbard, P. L., Bryant, C. L., & Tzedakis, P. C. (2009). Character of vegetational and environmental changes in southern Europe during the last glacial period; evidence from Lesvos Island, Greece. Quaternary Science Reviews, 28(13-14), 1317-1339. |
| 50 | Navarres | ACER | Carrión, José S; van Geel, Bas (1999): Fine-resolution Upper Weichselian and Holocene palynological record from Navarrés (Valencia, Spain) and a discussion about factors of Mediterranean forest succession. Review of Palaeobotany and Palynology, 106, 209-236. |
| 51 | Xinias | ACER | Bottema, S. (1979). Pollen analytical investigations in Thessaly (Greece). Palaeohistoria, 19-40. |
| 52 | Lake Van | Neotoma | Pickarski, N., & Litt, T. (2017). A new high-resolution pollen sequence at Lake Van, Turkey: insights into penultimate interglacial–glacial climate change on vegetation history. Climate of the Past, 13(6), 689-710. |
| 53 | Kupena | Neotoma | Tonkov, S., Lazarova, M., Bozilova, E., Ivanov, D., & Snowball, I. (2014). A 30,000-year pollen record from Mire Kupena, Western Rhodopes Mountains (south Bulgaria). Review of palaeobotany and palynology, 209, 41-51. |
| 54 | Lake Urmia | Neotoma | Djamali, M., de Beaulieu, J. L., Shah-hosseini, M., Andrieu-Ponel, V., Ponel, P., Amini, A., ... & Brewer, S. (2008). A late Pleistocene long pollen record from Lake Urmia, NW Iran. Quaternary Research, 69(3), 413-420. |
| 55 | Padul | - | Camuera, J., Jiménez-Moreno, G., Ramos-Román, M. J., García-Alix, A., Toney, J. L., Anderson, R. S., ... & Carrión, J. S. (2019). Vegetation and climate changes during the last two glacial-interglacial cycles in the western Mediterranean: a new long pollen record from Padul (southern Iberian Peninsula). Quaternary Science Reviews, 205, 86-105. |
| 56 | Sögüt Gölü | Neotoma | Zeist, W. van, Woldring, H., Stapert, D., 1975. Late quaternary vegetation and climate of southwestern Turkey. Palaeohistoria 53–143. |
| 57 | Dar Fatma | Neotoma | Stambouli-Essassi, S., Roche, E., & Bouzid, S. (2007). Evolution de la végétation et du climat dans le Nord-ouest. Geo-Eco-Trop, 31, 171-214. |
| 58 | Bajondillo | Neotoma | Cortés-Sánchez, M., Morales-Muñiz, A., Simón-Vallejo, M. D., Bergadà-Zapata, M. M., Delgado-Huertas, A., López-García, P., ... & Vera-Peláez, J. L. (2008). Palaeoenvironmental and cultural dynamics of the coast of Málaga (Andalusia, Spain) during the Upper Pleistocene and early Holocene. Quaternary Science Reviews, 27(23-24), 2176-2193. |
| 59 | Ghab | Neotoma | Bottema, S. (1987). Chronology and climatic phases in the near east from 16,000 to 10,000 BP. Chronologies in the Near East. Oxford: British Archaeological Reports, 295310. |
| 60 | Lake Zeribar | Neotoma | Van Zeist, W., & Bottema, S. (1977). Palynological investigations in western Iran. Palaeohistoria, 19-85. |
| 61 | Dead Sea | Neotoma | Miebach, A., Stolzenberger, S., Wacker, L., Hense, A., & Litt, T. (2019). A new Dead Sea pollen record reveals the last glacial paleoenvironment of the southern Levant. Quaternary Science Reviews, 214, 98-116. |





**Appendix B: Site availability per time window**

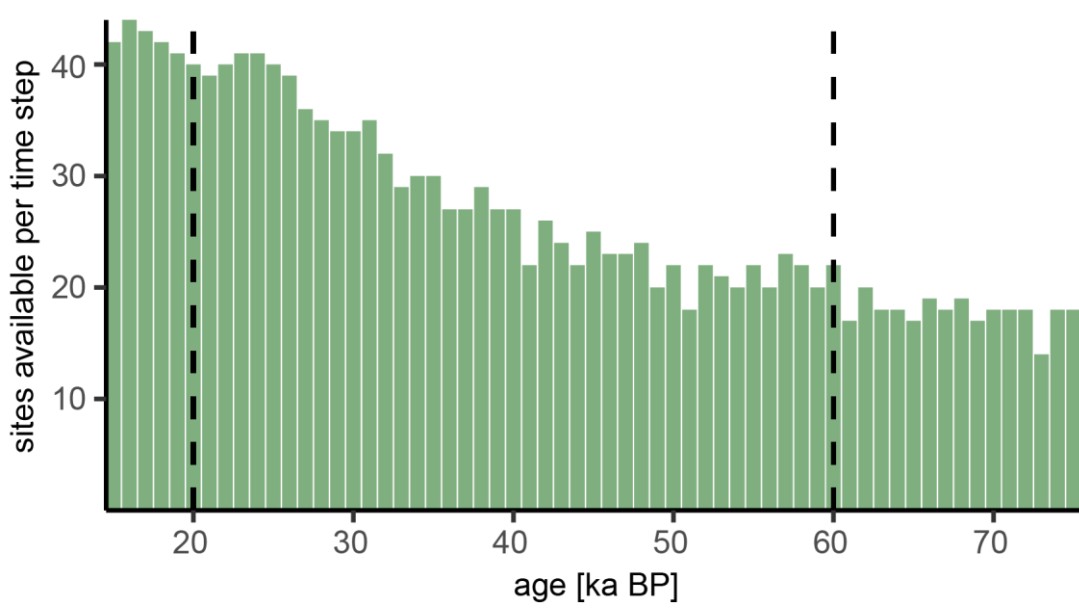


**Figure B1: Available sites with pollen data for each 1000 yr time window. The selection of sites was based on the availability of data between 60 and 20 ka BP (dashed lines).**

**Appendix C: Taxon classification and land-cover types (LCTs)**

**Table C1: Classification of plant taxa, genera, and families into land-cover types (LCTs), adapted from Githumbi et al.**
**(2022) for the Last Glacial period.**

| Land-cover types (LCTs) | Included plant taxa, genera, and families |
|---|---|
| Conifers | *Abies alba*, *Juniperus communis*, *Picea abies*, *Pinus* |
| Cold deciduous trees | *Alnus*, *Betula*, *Salix* |
| Temperate deciduous trees | *Acer*, *Buxus sempervirens*, *Carpinus betulus*, *Corylus avellana*, *Fagus sylvatica*, *Fraxinus excelsior*, *Populus*, *Quercus* deciduous (e.g., *Quercus cerris*, *Quercus robur*, *Quercus petraea*, *Quercus pubescens*), *Tilia*, *Ulmus* |
| Mediterranean trees | *Carpinus orientalis*, *Castanea sativa*, *Phillyrea*, *Pistacia*, *Quercus* evergreen (e.g., *Quercus coccifera*, *Quercus ilex*), *Sambucus nigra* |
| Total Trees | Conifers + Cold deciduous trees + Temperate deciduous trees + Mediterranean trees |
| Open land | Apiaceae, *Artemisia*, Asteraceae, Chenopodiaceae, Cyperaceae, Ericaceae, Fabaceae, *Filipendula*, *Plantago*, Poaceae, *Potentilla*, Ranunculaceae, Rubiaceae, *Rumex acetosa*, *Urtica* |





**Appendix D: Data reliability per taxon / LCT before and after data aggregation**

**Table D1: Comparison of the percentage of unreliable data for all taxa and LCTs before and after data aggregation into 1000 yr time bins.**

| Taxon / LCT | % unreliable data (aggregated) | % unreliable data | Taxon / LCT | % unreliable data (aggregated) | % unreliable data |
|---|---|---|---|---|---|
| **Conifers** | 19 | 19 | **Mediterranean trees** | 44 | 55 |
| *Abies alba* | 100 | 100 | *Carpinus orientalis* | 21 | 60 |
| *Juniperus communis* | 14 | 23 | *Castanea* | 63 | 69 |
| *Picea abies* | 21 | 22 | *Phillyrea* | 59 | 67 |
| *Pinus* | 3 | 4 | *Pistacia* | 39 | 41 |
| **Cold deciduous trees** | 17 | 26 | *Quercus* evergreen | 54 | 60 |
| *Alnus* | 48 | 62 | *Sambucus nigra* | 61 | 72 |
| *Betula* | 17 | 21 | **Open land** | 2 | 2 |
| *Salix* | 24 | 37 | Apiaceae | 12 | 21 |
| **Temp deciduous trees** | 19 | 27 | *Artemisia* | 5 | 7 |
| *Acer* | 53 | 71 | Asteraceae | 6 | 11 |
| *Buxus sempervirens* | 66 | 86 | Chenopodiaceae | 21 | 33 |
| *Carpinus betulus* | 47 | 56 | Cyperaceae | 8 | 11 |
| *Corylus avellana* | 26 | 44 | Ericaceae | 47 | 55 |
| *Fagus sylvatica* | 35 | 39 | *Fabaceae* | 46 | 52 |
| *Fraxinus excelsior* | 47 | 62 | Filipendula | 69 | 82 |
| *Populus* | 100 | 100 | *Plantago* | 34 | 51 |
| *Quercus* deciduous | 16 | 22 | Poaceae | 2 | 2 |
| *Tilia* | 49 | 62 | *Potentilla* | 67 | 79 |
| *Ulmus* | 40 | 49 | *Ranunculaceae* | 65 | 77 |
| | | | *Rubiaceae* | 71 | 90 |
| | | | *Rumex acetosa* | 69 | 83 |
| | | | *Urtica* | 95 | 99 |


**Author contributions**

OAK: Conceptualization, Data curation, Formal analysis, Investigation, Validation, Visualization, Writing – original draft preparation, Writing – review & editing. AM: Conceptualization, Supervision, Writing – review & editing. NV: Conceptualization, Project administration, Supervision, Writing – review & editing

**Competing interests**

The authors declare that they have no conflict of interest.




**Acknowledgments**

We thank Jon Camuera, Penélope González-Sampériz, Vaida Šeirienė, and Philipp Stojakowits for providing unpublished pollen data. Moreover, we express our sincerest gratitude to the people involved in managing and curating the ACER,
Neotoma, and PalyCZ pollen databases as well as all data contributors for providing their services and data to the public. We also thank Monika Doubrawa, Tilman Hartley, Johanna Hilpert, Philipp Schlüter and Isabell Schmidt for scientific discussions. The project "HESCOR" is receiving funding from the programme "Profilbildung 2022", an initiative of the Ministry of Culture and Science of the State of North-Rhine Westphalia, Germany (HESCOR PB22-081). The sole responsibility for the content of this publication lies with the authors.

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
