# Peer review of "Landscape reconstructions for Europe during the late Last Glacial (60–20 ka BP): A pollen-based REVEALS approach"

_Earth System Science Data, 2024_

## Author Comment (AC1)

ESSD-2024-306: Reply to comments from Referee #1

(Reviewer comments in **bold**, author responses in **blue**)

This study holds significant scientific and methodological value for reconstructing vegetation cover in Europe during the late Last Glacial period (60-20 ka BP). The authors applied the REVEALS approach based on numerous palynological records. An important methodological contribution of the work is the comparison between REVEALS results and palynological analysis data, along with attempts to explain the obtained discrepancies.

We would like to thank Referee #1 for the positive assessment of our manuscript and his/her constructive suggestions that help us to further strengthen our argumentation, data interpretation and increase to overall quality of the manuscript. Below we provide specific, point-to-point responses to all his/her comments and suggestions.

The authors present interesting data for stadials and interstadials of MIS 3 - an important yet controversial period of the Late Pleistocene. I found lacking the attempts to explain the obtained data for northeastern Europe, where unlike other regions, the reconstructed forest cover decreases during interstadial warming and humidification. What climatic mechanism could lead to such consequences? For instance, during the Holocene, warming and increased moisture typically result in northward forest expansion.

We thank R1 for highlighting this. We fully agree that we expect a higher tree cover during interstadials compared to stadials due to the higher temperatures and higher moisture availability. After having a closer look at the data itself, we came to the conclusion that e.g., the observed pattern of lower forest cover during early and mid-MIS 3 (Fig. 2, top right, ca 60–40 ka BP) is the result of the available data for this region during that time.

For some time intervals, very little pollen data are available from NE Europe (i.e., 3-5 records at times). Hence, the mean forest cover is substantially affected by the spatial distribution of those records. For example, the forest covers of Lake Yamozero (65°N, 70–40 ka BP) and Nesseltalgraben (Alpine region, 60-30 ka BP) are much more affected by their respective latitudes or proximity to ice sheets than by stadial-interstadial variability. As a consequence, the regional mean forest cover in NE Europe during mid-MIS 3 is heavily skewed towards lower forest covers as observed at Lake Yamozero and Nesseltalgraben. In turn, the 40–16 ka BP interval is skewed towards higher forest covers due to the inclusion of the data from the Carpathian Mountains (e.g. Jablunka and Safarka) and the exclusion of the Lake Yamozero data.

This effect is of particular importance for NE Europe due to the wide latitudinal range of the records as well as the generally low number of records available. We will clarify this in our manuscript and add an additional subfigure that highlights the data scarcity in certain regions that can lead to a bias in the mean forest cover percentages.

The paper repeatedly mentions that understanding vegetation variability during stadial-interstadial cycles and extreme glacial conditions is highly significant for demographic developments among Paleolithic hunter-gatherers, particularly regarding Neanderthal replacement by Homo sapiens. However, this aspect remains largely unexplored. Perhaps the authors have insights into how described natural conditions and climatic fluctuations influenced the development of human communities in Europe?

We thank Referee #1 for this comment and we fully agree that our discussion regarding the impact of vegetation dynamics on demographic developments of early humans is insufficient. In the revised manuscript, we will go into more detail on the ramifications of the observed vegetation dynamics and

how these might have affected the dispersal of Palaeolithic hunter-gatherers into Europe and the disappearance of Neanderthals.

**Line 316: South-West Asia? Why Asia?**

We appreciate the reviewer's comment. However, we are unable to locate the specific text mentioned at the indicated line. It's possible there was a mismatch in line numbering or manuscript version. We repeatedly refer to South-West Asia throughout the manuscript, because some of the pollen records included in our study are from Israel, Syria, eastern Turkiye, and Iran (see Fig. 1 in the original manuscript). We include these datasets in an attempt to gain a better picture of vegetation dynamics in Eastern Europe and the easternmost Mediterranean area – areas which otherwise lack datasets.

**Figs 3,4,5 B: What do the pollen percentages represent?**

We thank Referee #1 for highlighting that our plot labels lack clarity. All pollen percentages mentioned in Figs 3-5 B refer to the sum of "open land" taxa. We will change the figure captions accordingly to make this clearer to the reader.

---

## Author Response (AR1)

**Author's response to ESSD-2024-306**

We would like to thank the editor and the two anonymous reviewers time and efforts devoted to our manuscript. We are grateful for the constructive feedback and are convinced that they have further improved the quality of our manuscript. Below, we provide point-by-point answer to the reviewers' comments.

(Reviewer comments in **bold**, author responses in blue)

Reply to comments from Referee #1

This study holds significant scientific and methodological value for reconstructing vegetation cover in Europe during the late Last Glacial period (60-20 ka BP). The authors applied the REVEALS approach based on numerous palynological records. An important methodological contribution of the work is the comparison between REVEALS results and palynological analysis data, along with attempts to explain the obtained discrepancies.

We would like to thank Referee #1 for the positive assessment of our manuscript and his/her constructive suggestions that help us to further strengthen our argumentation, data interpretation and increase to overall quality of the manuscript. Below we provide specific, point-to-point responses to all his/her comments and suggestions.

The authors present interesting data for stadials and interstadials of MIS 3 - an important yet controversial period of the Late Pleistocene. I found lacking the attempts to explain the obtained data for northeastern Europe, where unlike other regions, the reconstructed forest cover decreases during interstadial warming and humidification. What climatic mechanism could lead to such consequences? For instance, during the Holocene, warming and increased moisture typically result in northward forest expansion.

We thank R1 for highlighting this. We fully agree that we expect a higher tree cover during interstadials compared to stadials due to the higher temperatures and higher moisture availability. After having a closer look at the data itself, we came to the conclusion that e.g., the observed pattern of lower forest cover during early and mid-MIS 3 (Fig. 2, top right, ca 60–40 ka BP) is the result of the available data for this region during that time.

In our revised manuscript, we added the following paragraph (lines 256–265) to correct this:

"For NE Europe, the opposite pattern emerges. It appears that during the coldest and driest intervals (MIS 4 and MIS 2) of the Last Glacial, tree populations remain at a moderate level and instead decline during the warmer and more humid period of MIS 3. However, we ascribe this pattern to data scarcity rather than climatic variability. Particularly in NE Europe, very few datasets are available (Fig. 2 B) and they span a large latitudinal range (from 46°N to 67°N). During mid-MIS 3, only limited data are available and thus the mean forest cover is strongly influenced by records from the high latitudes (e.g., Lake Yamozero, site 2, Fig. 1) or in close vicinity to ice sheets (e.g., Nesseltalgraben, site 21, Fig. 1), which tend to signal a very open landscape. During subsequent intervals (i.e., late MIS 3), more records from, e.g., the Carpathian Mountains, are included and thus shift the mean towards higher forest covers. Note that despite the overall site availability in NW Europe being similar or even lower compared to NE Europe (Fig. 2 B), these sites span a much smaller geographical area and climatic range. As a consequence, the presence or absence of individual sites has a reduced impact on the regional means."

Moreover, we now present the available sites per time step in Figure 2 B to further strengthen our argument and provide full transparency regarding the data used to create Figure 2 A, which our discussion is based on.

The paper repeatedly mentions that understanding vegetation variability during stadial-interstadial cycles and extreme glacial conditions is highly significant for demographic developments among Paleolithic hunter-gatherers, particularly regarding Neanderthal replacement by Homo sapiens. However, this aspect remains largely unexplored. Perhaps the authors have insights into how described natural conditions and climatic fluctuations influenced the development of human communities in Europe?

We thank Referee #1 for this comment and we fully agree that our discussion regarding the impact of vegetation dynamics on demographic developments of early humans is insufficient.

In Section 3 and 7 of our revised manuscript we discuss our data in the context of demographic developments of Palaeolithic hunter-gatherers:

Lines 276–278: "Also, shifts between stadial and interstadial conditions and their associated ecological gradients may have had considerable impact on the large-scale distribution of regional populations of hunter-gatherers during the Upper Palaeolithic (Maier et al., 2024).

Lines 326-341: "The arrival and spread of Palaeolithic hunter-gatherers between 43 to 40 ka BP in Europe (Shao et al., 2024) occurred during a phase of pronounced interstadials (ca. GI-12 to GI-9) and therefore relatively mild and humid climate conditions (Rasmussen et al., 2014). This climate state is also reflected in increased tree cover percentages, particularly in Eastern Europe (i.e., the Balkans and along the Danube), where climatically suitable east-west corridors may have temporarily opened up for human dispersal across Europe (Shao et al. 2024). Here, light woodland or a mosaic of open and forested vegetation was prevalent particularly in SE Europe (Fig. 2 A). Such ecotones provide ideal conditions for a high faunal biodiversity and are thus attractive habitats for hunter-gatherers. During stadials, the tree line shifted southwards and the landscape in the higher latitudes opened up. At the same time, phenological gradients related to the greening of the landscape in spring became more pronounced in the southern parts of Europe. The gradual cooling of interstadials likely led to a shift in largescale phenological patterns, presumably affecting the spatial distribution of populations and providing incentives to move, for instance, into the more southern areas of the Iberian Peninsula (Maier et al., 2024), or foster retreat to local niches (Timmermann, 2020). Combined with decreasing temperatures and moisture availability, much of Central and Eastern Europe might have become unfavourable for hunter-gatherers during pronounced stadials, such as Heinrich Event 4, and populations may have decreased (Shao et al. 2024). There are hints that Neanderthals had smaller mobility ranges than anatomically modern humans and occupied more fragmented habitats (Timmermann, 2020). Together, this may have fostered sensitivity to climate change during MIS 3 and its implications for ecosystems (Yaworsky et al., 2024), contributing in the long run to their disappearance.

Lines 483–489: "It has long been hypothesised to what extent climatic fluctuations have impacted these patterns of population dynamics on both temporal and spatial scales (Maier et al., 2022, 2024; Staubwasser et al., 2018; Timmermann and Friedrich, 2016). Here, corridors for the dispersal of Palaeolithic hunter-gatherers into different parts of Europe may have episodically opened up in relation to the opening of the landscape and shifts in ecological gradients. Furthermore, tree populations have persisted in local refugia in Central and Eastern Europe, where favourable environmental conditions could have provided shelter during subsequent stadial climatic downturns."

**Line 316: South-West Asia? Why Asia?**

We appreciate the reviewer's comment. However, we are unable to locate the specific text mentioned at the indicated line. It's possible there was a mismatch in line numbering or manuscript version. We repeatedly refer to South-West Asia throughout the manuscript, because some of the pollen records included in our study are from Israel, Syria, eastern Turkiye, and Iran (see Fig. 1 in the original manuscript). We include these datasets in an attempt to gain a better picture of vegetation dynamics in Eastern Europe and the easternmost Mediterranean area – areas which otherwise lack datasets.

**Figs 3,4,5 B: What do the pollen percentages represent?**

We thank Referee #1 for highlighting that our plot labels lack clarity. All pollen percentages mentioned in Figs 3-5 B refer to the sum of "open land" taxa.

We have modified the captions of Figs 3–5 in our revised manuscript (lines 295–296, 313–314, and 354–355) to enhance clarity.

**Reply to comments from Referee #2**

This work provides a landscape database reconstruction for the Late Pleistocene period (75,000-15,000 ka BP), online and free to use, using the already well used REVEALS approach. I found the background and methods well explained and clear; I particularly appreciated the part explaining why raw pollen data are not directly proportional to the vegetation cover of a given species. It is significant achievement and will likely be of large us in different scientific communities.

We thank Referee #2 for her/his very positive assessment and we appreciate the valuable input provided on our work.

1-/ Line 304: « The stadial-interstadial variability is primarily characterised by an increase in openland percentages. » this sentences is unclear: it could be an increase or a decrease depending on the period (stadial or interstadial. Maybe something like « The stadial-interstadial variability is primarily characterised by a change in open-land percentages. » or be more specific as to the succession of changes from stadial to interstadial.

We agree with Referee #2 that the phrasing is ambiguous. Hence, we have rephrased the paragraph (lines 316–324) to be clearer and less prone to misinterpretation:

"Stadial-interstadial variability is primarily characterized by an increase in open-land percentages during stadials and a decrease in open-land percentages during interstadials. Such increases in open-land percentages have led to a southward displacement of the tree line in western Central Europe, while limited tree population in eastern Central Europe persist. During interstadials, decreases in open-land percentages are accompanied with a northward expansion of the tree line. These observations are in line with the interpretation of proxy records across Europe (Fletcher et al., 2010; Landais et al., 2022, Tzedakis et al., 2013). However, the scarcity of available pollen records for GI-9 from Northern and North-eastern Europe substantially hinders our ability for further inquiry. Additional pollen records are required to investigate the spatio-temporal framework of tree-line recession in a north-easterly direction during interstadials in more detail."

2-/ Line 305: Is it really northward? My understanding is that it should be southward as the high latitudes are mainly covered by herbaceous. The tree-line shift to higher latitudes must have been with trees on south and herbaceous on north. Please clarify.

We thank Referee #2 for pointing out this mistake. As mentioned above, we have rephrased the entire paragraph (lines 316–324) to make it clearer to the readers and have also corrected this mistake.

---

## Author Response (AR2)

**Author's response to ESSD-2024-306**

We would like to thank the editor for their constructive feedback and their diligent review of our manuscript. Below, we provide point-by-point answer to the editors comments.

(Editor comments in **bold**, author responses in blue)

Reply to comments from the Editor

Chapter 6 Data availability: could you provide in this chapter the direct link to your data Kern et al. with DOI (https://doi.pangaea.de/10.1594/PANGAEA.973049, the link in the manuscript only leads to the PANGAEA landing page)

Thank you for pointing out this mistake. We have corrected the link to the correct PANGAEA dataset.

Page 4 instead of the citation of the PANGAEA repository with the description of the repository by Felden et al. 2023, could you list here all the PANGAEA data publications that you used and cite the PANGAEA data publications also in the reference list, (all data publications that you re-used should appear in the reference list of your manuscript).

We have changed the citation and included citations for all datasets that are available on PANGAEA that were used in this study. The citations have been added to the references accordingly.

As for the ESSD manuscript of Schild et al. 2025 LegacyVegetation: Northern Hemisphere reconstruction of past plant cover and total tree cover from pollen archives of the last 14 kyr. Earth System Science Data, 17(5), 2007-2033, https://doi.org/10.5194/essd-17-2007-2025

we were asked by NEOTOMA to assure the traceability of the NEOTOMA source of the published data showing the NEOTOMA DOIs - e.g. Schild et al added the information of the NEOTOMA DOI in their data publication LegacyPollen2.0, an updated global taxonomically and temporally standardized fossil pollen dataset of 3728 palynological records https://doi.org/10.1594/PANGAEA.965907, Li et al., 2024 and finally discussed in the revised manuscript the reconstructed Northern hemisphere data reconstruction (not the gobal reconstruction): Herzschuh, U., Ewald, P., Schild, L., Li, C., and Böhmer, T.: LegacyVegetation: Northern Hemisphere reconstruction of past plant cover and total tree cover from pollen archives of the last 14 ka, PANGAEA [data set], https://doi.org/10.1594/PANGAEA.974798, 2025

(Eventually these PANGAEA data sets are also interesting for you to cite in your ESSD manuscript and reference list? This part is not meant as editorial requirement, only to be considered, if it is of use for your ESSD manuscript).

Thank you for highlighting these very interesting datasets. Unfortunately, however, they both cover younger time periods that only marginally overlap with the time interval of our study. Otherwise we would have gladly used these existing databases in our study. Nonetheless, we do cite the original publications in our study as they are highly relevant from a methodological perspective.

The solution in your case it that you could provide the NEOTOMA DOI for the NEOTOMA sources you used, e.g. in the form of a new Zenodo publication on your NEOTOMA metadata. You then can link

in Zenodo your new Zenodo NEOTOMA DOI metadata publication to your PANGAEA data publication Kern et al. 2024 https://doi.pangaea.de/10.1594/PANGAEA.973049, and ask PANGAEA to link your data publication to your new Zenodo metadata publication as related data publication, that usually can be done very fast in PANGAEA and is easily added to existing DOI data publications. During this process, please also ask PANGAEA to link your Code publications in Zenodo in form of 'related code / software' within your PANGAEA data publication.

Thank you very much for providing such an in-depth and step-by-step solution. We have uploaded the metadata with all NEOTOMA DOIs to Zenodo (<a href="https://doi.org/10.5281/zenodo.16812766">https://doi.org/10.5281/zenodo.16812766</a>) and have contacted PANGAEA to link the PANGAEA datasets to this Zenodo publication, as well as the others included in this study (<a href="https://doi.org/10.5281/zenodo.12782121">https://doi.org/10.5281/zenodo.12625222</a> (REVEALS R code)). These Zenodo publications have been added to the PANGAEA dataset (<a href="https://doi.pangaea.de/10.1594/PANGAEA.973049">https://doi.pangaea.de/10.1594/PANGAEA.973049</a>) under the "related to:" section and have also been added to the manuscript as well as the references.